# The splicing factor XAB2 interacts with ERCC1-XPF and XPG for R-loop processing

Evi Goulielmaki[1,6], Maria Tsekrekou [1,2,6], Nikos Batsiotos[1,2], Mariana Ascensão-Ferreira [3], Eleftheria Ledaki[1], Kalliopi Stratigi [1], Georgia Chatzinikolaou[1], Pantelis Topalis [1], Theodore Kosteas[1], Janine Altmüller[4], Jeroen A. Demmers [5], Nuno L. Barbosa-Morais [3] & George A. Garinis [1,2✉]

RNA splicing, transcription and the DNA damage response are intriguingly linked in mammals but the underlying mechanisms remain poorly understood. Using an in vivo biotinylation tagging approach in mice, we show that the splicing factor XAB2 interacts with the core spliceosome and that it binds to spliceosomal U4 and U6 snRNAs and pre-mRNAs in developing livers. XAB2 depletion leads to aberrant intron retention, R-loop formation and DNA damage in cells. Studies in illudin S-treated cells and $Csb^{m/m}$ developing livers reveal that transcription-blocking DNA lesions trigger the release of XAB2 from all RNA targets tested. Immunoprecipitation studies reveal that XAB2 interacts with ERCC1-XPF and XPG endonucleases outside nucleotide excision repair and that the trimeric protein complex binds RNA:DNA hybrids under conditions that favor the formation of R-loops. Thus, XAB2 functionally links the spliceosomal response to DNA damage with R-loop processing with important ramifications for transcription-coupled DNA repair disorders.

[1] Institute of Molecular Biology and Biotechnology, Foundation for Research and Technology-Hellas, Heraklion, Crete, Greece. [2] Department of Biology, University of Crete, Heraklion, Crete, Greece. [3] Instituto de Medicina Molecular João Lobo Antunes, Faculdade de Medicina da Universidade de Lisboa, Avenida Professor Egas Moniz, Lisboa, Portugal. [4] Cologne Center for Genomics (CCG), Institute for Genetics, University of Cologne, Cologne, Germany. [5] Proteomics Center, Netherlands Proteomics Center, and Department of Biochemistry, Erasmus University Medical Center, Erasmus, the Netherlands. [6] These authors contributed equally: Evi Goulielmaki, Maria Tsekrekou. ✉email: garinis@imbb.forth.gr

The spliceosome is a highly dynamic, ribonucleoprotein complex that comprises five small nuclear (sn) RNAs (the U1, U2, U4, U5, and U6 snRNAs) and a growing number of associated splicing factors that enable the selective intron excision of nascent pre-mRNA transcripts prior to mRNA translation[1]. Splicing initiates with the binding of the U1 snRNP to the GU sequence at the 5′ splice site of an intron and the zinc finger protein splicing factor 1 (SF1) that binds to the intron branch point sequence. The U2 snRNP displaces SF1 and binds to the branch point sequence followed by ATP hydrolysis. The snRNPs U2 and U4/U6 position the 5′ end and the branch point in proximity. The latter stimulates snRNP U5 and the 3′ end of the intron is brought into proximity and joined to the 5′ end. The resulting lariat is released with U2, U5, and U6 bound to it[2]. Pre-mRNA splicing is functionally coupled to transcription[3,4]. The process of mRNA synthesis is known to regulate constitutive and alternative splicing (AS) through the physical association of splicing factors to the elongating RNAPII[5], the 3D chromatin structure[6,7] or through the fine-tuning of RNA polymerase II (RNAPII) elongation rates[8]. Recent findings support the notion that DNA damage is inherently linked to splicing[9–11]. The presence of irreparable DNA lesions may influence RNA splicing through the selective nuclear transport of splicing factors[12,13], the inhibition of the elongating RNAPII[9], the interaction of RNAPII- and spliceosome-associated factors[14] or the displacement of the spliceosome when RNAPII is stalled at DNA damage sites[11].

XPA-binding protein (XAB)-2 is the human homologue of the yeast pre-mRNA splicing factor Syf1 that contains 15 half a tetra-tricopeptide (HAT) repeats shared by proteins involved in RNA processing[15,16]. Disruption of the *Xab2* gene results in pre-implantation lethality in mice[17]. Thus, there are currently no available data to address the functional role of XAB2 in transcript maturation and the DNA damage response (DDR) in vivo. Using a yeast two-hybrid screen and in vitro pull-down assays, XAB2 was shown to interact with the nucleotide excision repair (NER) factors XPA, CSA and CSB proteins and with RNAPII[18]. The yeast homolog of XAB2, Syf1 was also shown to be a TREX (transcription and export complex)-interacting factor[19]. Microinjection of antibodies raised against XAB2 inhibited the recovery of RNA synthesis after UV irradiation and normal RNA synthesis in fibroblasts[18]. Follow-up work showed that XAB2 interacts with PRP19[20] an ubiquitin-protein ligase involved in pre-mRNA splicing and DNA repair[21,22] and Aquarius (AQR) an intron-binding spliceosomal factor that links pre-mRNA splicing to small nucleolar ribonucleoprotein biogenesis[23]. Moreover, XAB2 is functionally involved in homology-directed repair and single-strand annealing[24] and it was recently shown to be directly involved in pre-mRNA splicing of many genes in vitro, including POLR2A[25].

To dissect the functional contribution of XAB2 in mRNA synthesis and DNA repair, we established an in vivo biotinylation tagging approach in mice and primary cells. Our findings provide evidence that XAB2 is essential for NER, the pre-mRNA splicing and for R-loop processing. We show that XAB2 is part of a core spliceosome complex that associates with the UsnRNAs and that transcription-blocking DNA damage triggers the release of XAB2 from all RNA targets tested. RNAi-mediated knockdown of XAB2 leads to decreased RNA synthesis, aberrant intron retention, R-loop accumulation and DNA damage. A series of immunoprecipitation strategies revealed that XAB2 interacts with ERCC1-XPF and XPG endonucleases and that the XAB2 complex is recruited to RNA:DNA hybrids under conditions that favor the accumulation of R-loops. We propose that XAB2 functionally links persistent DNA damage with the core spliceosome and the processing of R-loops highlighting the causal contribution of transcription-blocking DNA lesions to the progeroid and developmental defects associated with TC-NER disorders[26–29].

## Results

**Generation of biotin-tagged XAB2 mice**. Ablation of *Xab2* in the murine germline results in preimplantation lethality, indicating an essential role for XAB2 during mouse embryogenesis[17]. To delineate the functional role of XAB2, we generated knock-in animals expressing XAB2 fused with a tandem affinity purification tag comprising two affinity moieties, a 1X FLAG tag and a 15aa Avitag sequence, separated by a Tobacco Etch Virus (TEV) site for easy tag removal. The tag was inserted before the stop codon of the last exon 19 (Fig. 1a). The targeting vector was transfected to 129/SV ES cells expressing the Protamine 1-Cre recombinase transgene, which efficiently excises the neomycin cassette in the male germ line[30]. After selecting properly targeted clones (Fig. 1b; as indicated), two independently transfected clones were used to generate germ line-transmitting chimeras. Avitag-fused heterozygous males (*aviXab2*[+/−] mice) were back-crossed and maintained in a C57/BL6 background. Homozygous *aviXab2*[+/+] knock-in animals were then crossed with mice ubiquitously expressing the HA-tagged BirA biotin ligase transgene[31]. BirA is a bacterial ligase that specifically recognizes and successfully biotinylates the short 15aa biotinylation Avitag, thus creating a high affinity "handle" for the in vivo isolation of XAB2-bound protein complexes in aviXab2[+/+]; birA (from now on designated as bXAB2) mice by binding to streptavidin. Importantly, biotinylation of the inserted Avitag sequence does not interfere with murine development as bXAB2 animals are born at the expected Mendelian frequency (Fig. 1c), grow normally (Fig. 1d), and show no developmental defects or other pathological features (Fig. 1e). Nuclear extracts of P15 livers from aviXAB2 or bXAB2 mice were probed with streptavidin-HRP (stp-HRP) confirming the biotinylation of bXAB2 in vivo (Fig. 1f). The use of anti-FLAG and anti-HA confirmed the presence of the knock-in allele and the BirA transgene, respectively. Pulldowns with 0.6 mg of nuclear extracts from livers of P15 bXAB2[+/+] mice revealed the percentage of biotinylated XAB2 in the pulldown (90%) and flow through (fth; 10%) fraction (Fig. 1g). Previous findings suggest that XAB2 is involved in the transcription-coupled repair (TC-NER) sub-pathway of NER[18] and during the end resection step of homologous recombination repair[24]. To test whether the Avitag sequence interferes with the putative function of XAB2 in DNA repair in vivo, we further exposed mouse embryonic fibroblasts (MEFs) to ultraviolet irradiation (UV) as well as to mitomycin C (MMC). MMC is a potent genotoxic agent which inhibits replication and transcription of DNA by introducing DNA inter-strand cross-links that prevent dissociation of the strands. We find a similar percentage of UV-induced unscheduled DNA synthesis in bXAB2 cells compared to wild-type (wt.) controls (Fig. 1h) and that bXAB2 MEFs are not hypersensitive to UV irradiation (Fig. 1i) or to MMC (Fig. 1j). Thus, bXAB2 animals develop normally to adulthood and are proficient at repairing UV-induced cyclobutane pyrimidine dimers (CPDs) and DNA interstrand cross-links (ICLs).

**A proteomics strategy reveals XAB2-bound protein partners involved in RNA processing and DNA repair**. To isolate and characterize XAB2-associated protein complexes during postnatal hepatic development, we combined the in vivo biotinylation tagging approach[31] with a hypothesis-free, high-throughput proteomics strategy. To do this, we prepared nuclear extracts from livers of P15 bXAB2 and BirA mice using high-salt extraction conditions. Nuclear extracts from BirA mice were used to identify proteins non-specifically biotinylated by the BirA ligase (Fig. 2a). The liver is a relatively homogeneous organ that accurately depicts the growth status of the developing animal.

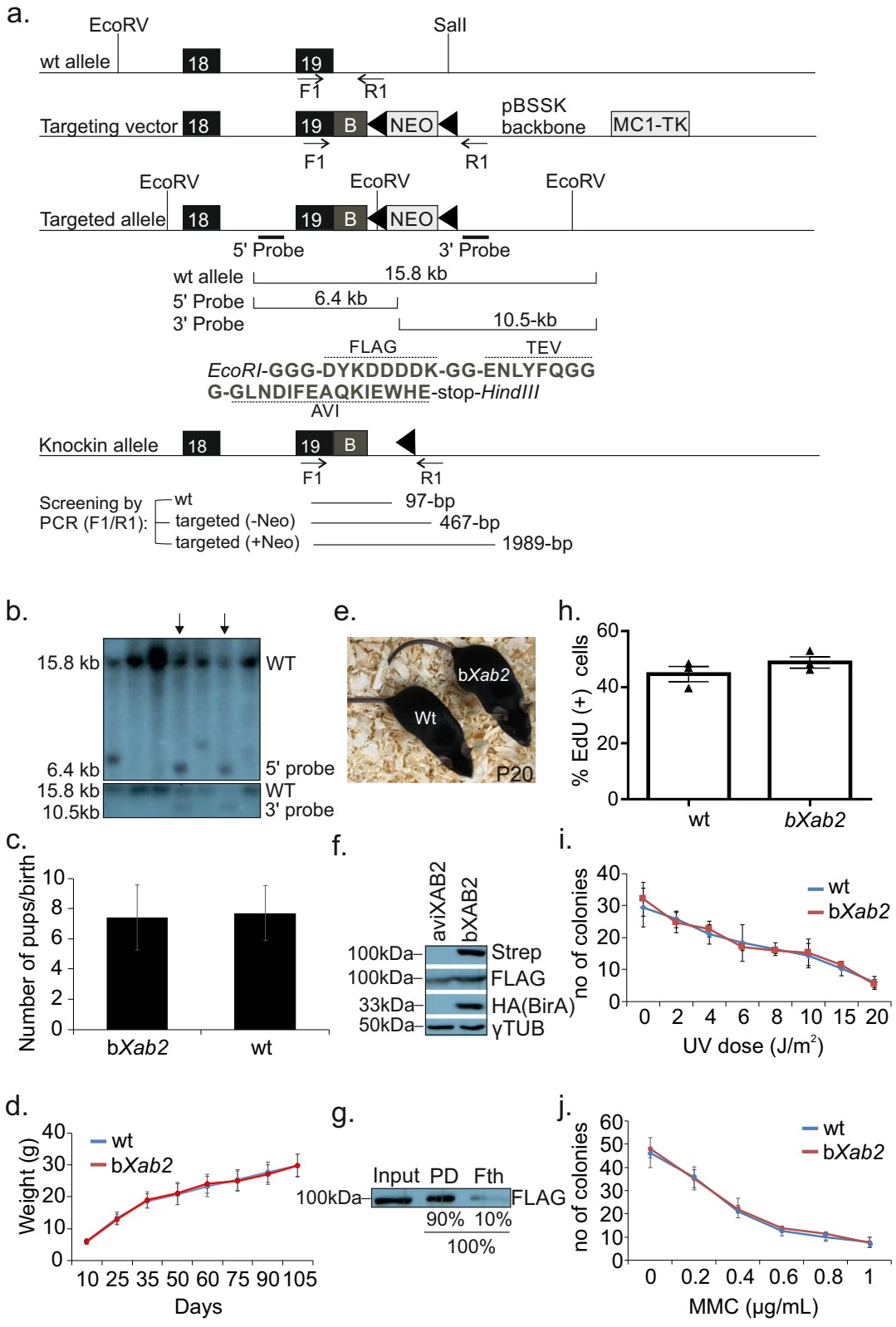

Nuclear extracts were treated with benzonase and RNase A, to ensure that DNA or RNA does not mediate the identified protein interactions. Nuclear extracts were further incubated with streptavidin-coated beads and bound proteins were eluted. To confirm the successful pulldown of XAB2, eluates were subjected to Western blotting for FLAG and for the pre-mRNA-processing factor 19 (PRP19), a known interactor of XAB2[32] (Fig. 2b). Next,

the proteome was separated by 1D SDS-PAGE (Fig. 2c) (~12 fractions) followed by in-gel digestion and peptides were analyzed with high-resolution liquid chromatography-tandem mass spectrometry (nLC MS/MS) on a hybrid linear ion trap Orbitrap instrument. From three biological replicates, which comprised a total of 72 MS runs, we identified a total of 1167 proteins (Supplementary Data 1) with 636 proteins (54.49%) shared between

**Fig. 1 Generation of biotin-tagged XAB2 animals. a** Schematic representation of knock-in mice expressing XAB2 fused with the 1xFLAG tag, a 15aa Avi tag sequence and a Tobacco Etch Virus (TEV) site separating the two tags. EcoRI and HindIII sites are synthetic; **b**: Flag-Tev-AVI fragment. **b** Two independently transfected ES clones (marked with arrow heads) were used to generate germline transmitting chimeras that were backcrossed with C57Bl/6 mice to generate $aviXab2^{+/-}$ pups. Homozygous $aviXab2^{+/+}$ knock-in animals (aviXAB2) were then crossed with mice ubiquitously expressing the HA-tagged BirA biotin ligase transgene ($aviXab2^{+/+}$;birA, designated as bXAB2). **c** Average number of wt ($n = 185$ pups/24 birth events) and bXAB2 ($n = 319$ pups/44 birth events) pups born per birth event. **d** Weight (grams; g) of wt. and bXAB2 animals ($n = 8$) at the indicated time points. **e** A photograph of P20 wt. and bXAB2 animals. **f** In vivo biotinylation of the short 15aa Avi-tag in bXAB2 animals. Nuclear extracts from P15 livers of mice expressing either only aviXAB2 (XAB2) or aviXAB2 and BirA biotin ligase (bXAB2) were tested by Western blot. The blot was probed with streptavidin-HRP (stp-HRP), anti-FLAG and anti-HA. **g** Biotinylation efficiency in bXAB2 livers. The percentage of biotinylated XAB2 in the pulldown (PD) and (90%) and flow through (Fth) fraction (10%; fth) was calculated by performing pull-down with 0.7 mg of nuclear extract derived from 15-day old bXAB2$^{+/+}$;BirA livers and M-280 paramagnetic beads in excess. **h** % of EdU (+) MEFs derived from bXAB2 ($n = 324$ cells examined /3 independent experiments) and wt. mice ($n = 480$ cells examined/3 independent experiments) 2.5 h after UV irradiation. **i** Survival of primary bXAB2 and wt. MEFs to UV ($n = 3$ per time point) or (**j**). MMC at the indicated doses ($n = 3$ per dose point). The images shown on Fig. 1f and g are representative of experiments that were repeated three times. Error bars indicate SEM. among $n = 3$ biological replicates, unless otherwise stated. All scatter and bar blots in this manuscript are presented as mean ± SEM. P values were calculated by two-tailed Student's t test. Source data provided as a Source Data file.

all three measurements under stringent selection criteria (Fig. 2d, Supplementary Data 2; see STAR Methods). To functionally characterize this dataset, we next subjected the 636-shared XAB2-bound proteins to gene ontology (GO) classification. Those biological pathways (Fig. 2e) or processes (Fig. S1A) containing a significantly disproportionate number of proteins relative to the murine proteome were flagged as significantly over-represented (FDR < 0.05). At this confidence level, the over-represented biological processes and pathways involved 372 out of the initial 636 XAB2-bound core proteins; the latter set of proteins also showed a significantly higher number of known protein interactions (i.e., 958 interactions) than expected by chance (i.e., 531 interactions; Fig. 2f) indicating a functionally relevant and highly interconnected protein network. Using this dataset, we were able to discern four major XAB2-associated protein complexes involved in (i) pre-mRNA splicing ($p \leq 3.2 \times 10^{-38}$, e.g., AQR, BCAS2, PRP19, PRP8, SNRP40), (ii) RNA transport ($p \leq 1.8 \times 10^{-7}$, e.g., NUP107, NUP133, NUP153, NUP160, NUP205, NUP210, NUP50, NUPl2), (iii) Cell cycle ($p \leq 7.1 \times 10^{-6}$, e.g., ANAPC1, ANAPC 2, ANAPC 4, ANAPC 5, ANAPC 7, CDC23, CDC27, FZR1, RAD21, SMC1a, SMC3, STAG1, STAG2) and iv. Ribosome biogenesis ($p \leq 4.8 \times 10^{-6}$, e.g., BMS1, GNL3, MDN1, NOP58, UTP15, WDR36, WDR43, WDR75, XRN2). Together, these findings indicate that the great majority of XAB2-bound protein partners are functionally involved in RNA processing and genome utilization processes. The role of XAB2 in splicing in vitro is already well-documented. In support, spliceosomal factors were highly enriched in the bXAB2-bound proteome of P15 livers. Nevertheless, as splicing factors are often non-specifically enriched in affinity enrichment-mass spectrometry approaches[33], we first challenged the specificity of XAB2 interaction with the pre-mRNA splicing machinery by comparing the bXAB2 MS liver proteome to that derived previously from P15 biotin-tagged xeroderma pigmentosum complementation F (bXPF) livers under similar experimental conditions[22]. XPF is the obligatory interacting partner in the NER structure-specific ERCC1-XPF endonuclease complex[34]. Unlike bXPF, we find that bXAB2 interacts with several core spliceosome factors and more than half of them are proteins involved in the PRP19 and PRP19-related complex (nine out of 16 pre-mRNA splicing factors) (Fig. 2g; colored in red).

**Depletion of XAB2 leads to defective NER.** Previous findings revealed that XAB2 is involved in the transcription-coupled sub-pathway of NER (TC-NER)[18,33]. To test for the functional role of XAB2 in NER, we carried a series of pulldown assays in nuclear extracts of UV-irradiated (10 J/m$^2$) and control bXAB2 primary MEFs. We find that bXAB2 interacts with the DNA damage-binding protein-1 (DDB1) involved in UV-induced DNA damage recognition[34], and with a portion of xeroderma pigmentosum, complementation group A protein (XPA) known to assemble the NER incision complex at sites of DNA damage[35], in control and UV-irradiated cells (Fig. S1B). Confocal microscopy in UV-irradiated HEPA cells revealed that XAB2 does not co-localize with the UV-induced CPDs (Fig. S1C). The great majority of XAB2 is distributed throughout the nucleoplasm but is excluded from the denser DAPI-stained heterochromatin regions or the nucleolus. In line with previous findings[24], we also find that XAB2 forms punctate staining, that is distinctively adjacent to the DNA damage marker γH2Ax (Fig. S1D). Knockdown of XAB2 in siXab2 HEPA cells (Fig. S1E) results in increased cell death (Fig. S1F), reduced proliferative capacity (Fig. S1G) at 72 h post-transfection, and in the presence of irreparable UV-induced CPDs at 24 h post-UV irradiation (Fig. 3a). The role of XAB2 in transcription-coupled repair has been previously documented[18]. In the present work, we find that siXab2 cells manifest a noticeable decrease in unscheduled DNA synthesis at 2.5 h post-UV irradiation (measuring global genome NER; Fig. 3b)[36]. siXab2 cells also present with substantial delay in the recovery of RNA synthesis (assessing TC-NER; Fig. 3c) as evaluated by the fluorescent detection of bromouridine (BrU) labeled nascent RNA at 4 h post-UV irradiation. In agreement with previous findings[25], knockdown of XAB2 reduced RNAPII (Fig. S1H) but did not affect the protein levels of NER factors DDB1, XPA and ERCC1, (Fig. S1I). Next, we investigated whether DDB1 and XPA are efficiently recruited at DNA damage sites in UV-irradiated siXab2 HEPA cells. We find that DDB1 co-localizes competently with UV-induced CPDs in siXab2 cells (Fig. 3d). In contrast, XPA fails to form visible foci and properly recruit on UV-induced CPDs in these cells (Fig. 3e). Finally, we find that treatment with Iso-Ginkgetin (IsoG), a pre-mRNA splicing inhibitor leads to severe splicing defects as documented by the accumulation of pre-mRNAs and the decrease of corresponding mRNA levels, (Fig. S1J–K) but has no significant impact on the repair of UV-induced CPDs in HEPA cells indicating that splicing per se does not affect NER.

**XAB2 is part of a core spliceosome complex that binds on UsnRNAs.** Follow-up pulldown experiments confirmed the mass spectrometry interactions and revealed that the endogenous bXAB2 interacts with AQR, PRP19, and BCAS2, an integral component of the spliceosome that is required for activating pre-mRNA splicing[37] in P15 livers (Fig. 4a). The interaction of bXAB2 with core spliceosome factors in P15 livers prompted us to test whether XAB2 is in complex with RNA in vivo. To do so, we carried out a series of RNA immunoprecipitation (RIP)

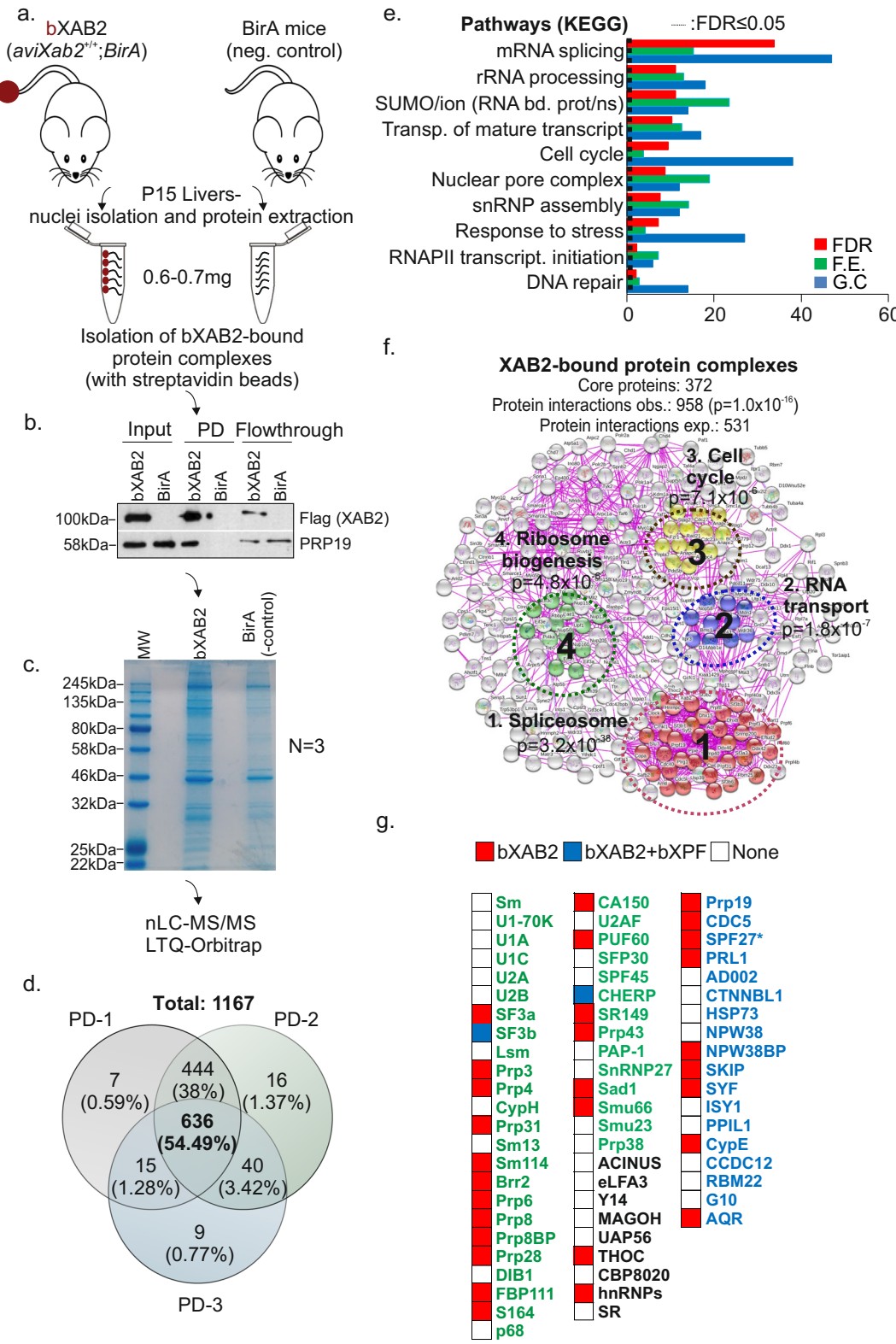

experiments in P15 bXAB2 liver nuclear extracts incubated with streptavidin-coated beads (for bXAB2). The co-precipitated RNA was eluted, reversely transcribed, and quantified for spliceosomal UsnRNAs by quantitative (q)PCR. We find that bXAB2 associates with U4, U5 and U6 small nuclear RNAs (snRNAs) but shows low to no specificity for U1, U2, or U3 snRNA (Fig. 4b). Unlike with XAB2 and AQR, RIP assays revealed that BCAS2 is not recruited to U4 and U6 snRNAs (Fig. 4c). To test whether XAB2

binds directly to UsnRNAs, we performed a series of gel mobility shift assays using a biotinylated RNA oligo that was designed based on the sequence similarities of U4, U5, and U6 snRNAs with the minimal RNA binding sequence of HCF107, a chloroplast-localized protein with repeats of HAT motifs (Fig. S1L). This strategy reveals that the XAB2 IP eluent from nuclear extracts of MEFs consumes the RNA oligo but not its mutant version (Fig. S1M) indicating that the XAB2 complex

**Fig. 2 XAB2 interacts with protein complexes involved in transcript maturation. a** Schematic representation of the isolation of bXAB2 complexes derived from livers of P15 bXAB2 animals expressing the BirA transgene and BirA transgenic control mice. **b** bXAB2 pulldowns (PD) and western blot with anti-FLAG (for XAB2) and anti-PRP19 in nuclear extracts derived from P15 bXab2 and BirA livers ($n = 3$ livers). **c** A representative SDS-PAGE gel of proteins extracts derived from P15 XAB2 and BirA transgenic mice. **d** Venn's diagram of bXAB2-bound protein factors from three independent pull-downs (PD) and subsequent MS analyses; % depicts the percentage of XAB2-bound proteins involved in the indicated biological process over the total number of XAB2-bound proteins (**e**). List of significantly over-represented pathways (KEGG). The −log(FDR: False Detection Rate), which is calculated by Fisher's exact test right-tailed (one-sided), sorts the biological processes. The red-dotted line marks the threshold of significance at FDR ≤ 0.05. Count (G.C.): the number of identified XAB2-bound protein factors involved in the indicated biological process. F.E. (Fold Enrichment), represents the ratio of XAB2-bound proteins involved in a process (sample frequency) to the total genes involved in the process (background frequency). **f** Number of observed (obs.) and expected (exp.) known protein interactions within the core 372 XAB2-bound protein set and schematic representation of the four significantly over-represented XAB2-bound protein complexes based on experimental evidence. **g** Heat map representation of bXAB2- and bXPF-bound proteins involved in the core spliceosome in P15 bXAB2 and bXPF livers. Constitutive factors of snRNPs are green-colored; the PRP19 and the PRP19-related complex are blue-colored. Protein components of the TREX and exon junction complex are black-colored. SPF27 (BCAS2) is marked with an "*" as it has been identified in two out of the 3 MS runs in P15 bXAB2 livers but has been further confirmed in downstream immunoprecipitation/pulldown assays. The images shown in Fig. 2b and c are representative of experiments that were repeated three times. All scatter and bar blots in this manuscript are presented as mean ± SEM. P values were calculated by two-tailed Student's t test, unless otherwise indicated. Source data provided as a Source Data file.

preferentially binds to the UsnRNA-specific sequence. Next, to test whether XAB2 binds directly to the same RNA sequence, we incubated increasing amounts of the recombinant hnXAB2 with the biotinylated RNA oligo (Fig. S1N). This approach revealed no consumption of the RNA oligo. Thus, XAB2 does not bind directly to but is in complex with UsnRNAs likely due to its interaction with, e.g., pre-mRNA-splicing factors. RNA-Seq analysis in siXab2 HEPA cells and mESCs 48 h post-transfection revealed 255 differentially expressed genes (124 genes were upregulated; 131 downregulated genes; fold change: ±1.2, q-value ≤ 0.05) (Supplementary Data 3) in siXab2 HEPA cells and 333 differentially expressed genes (41 upregulated and 291 downregulated; fold change: ±1.2, q-value ≤ 0.05) (Supplementary Data 4) in siXab2 mESCs, with a striking over-representation of ribosomal protein genes. Using this dataset, we next tested whether knockdown of Xab2 has an effect on splicing decisions, i.e., exon skipping (ES), intron retention (IR), and alternative selection of 5′ or 3′ splice sites (Fig. 4d). Of the 126335 and the 56677 alternative spliced (AS) events detected in HEPA cells and mESCs respectively, 774 and 1134 were found differentially spliced in siXab2 cells (probability of differential splicing >0.8 and an absolute change in percent spliced-in (ΔPSI) >5%). We find that the great majority (i.e., 805 out of 849, 94.82%, $p = 10^{-16}$) of IR events with a significant probability of differential splicing are retained in siXab2 HEPA and mESCs cells further confirming previous findings in HeLa cells[25]. Moreover 58.03% of alternative exons (i.e., 430 out of 741, $p = 10^{-5}$) are skipped when Xab2 is ablated in HEPA cells, indicating a global impairment of the splicing machinery in siXab2 cells (Fig. 4e–h, Fig. S2A–J, Supplementary Data 5, and Supplementary Data 6).

To investigate which biological processes are mostly affected by XAB2 knockdown-mediated AS, we subjected the AS events found to GO classification. At the confidence level set in our analysis (FDR ≤ 0.05), we find that biological processes related, among others, to cell cycle, RNAPII-mediated transcription, DNA repair, and RNA processing comprise a disproportional number of transcripts in both mESCs and HEPA cells relative to the murine transcriptome (Figs. S3A, S3B). RIP in P15 livers revealed that XAB2 binds to the pre-mRNAs of several genes involved in e.g. DNA repair and the DDR (Fig. S3C) as well as to the pre-mRNAs of genes known to be highly transcribed during postnatal hepatic development (e.g., Ghr, Igf1) (Fig. S3D). Transcripts with retained introns often contain premature stop codons (PTCs), which leads to their cytoplasmic degradation by the nonsense-mediated RNA decay machinery[38,39] or, as in the case of XAB2-mediated intron retained transcripts to aberrant RNA-decay pathways[25]. Otherwise, a subset of these transcripts

may be stably retained within the nucleus for later processing[40]. Beginning at 48 h post-transfection, we find that the pre-mRNA levels of transcripts with retained introns gradually accumulate in HEPA cells depleted for XAB2 with two distinct siRNAs (Fig. 4i and Fig. S3E) whereas their corresponding mRNA levels progressively decrease (Fig. 4j) indicating a defect in splicing rather than IR regulation in siXab2 cells. Thus, XAB2 is part of the core spliceosome complex and has a functional role in pre-mRNA splicing; it is recruited on UsnRNAs and pre-mRNAs and it is required for proper mRNA splicing.

**XAB2 is released from UsnRNAs and pre-mRNAs upon DNA damage and transcription blockage.** UV-induced DNA lesions trigger the displacement of co-transcriptional mature catalytically active spliceosome (U2, U5, and U6)[11]. In agreement, we find that the chromatin association of U4 and U6 snRNAs is reduced 2 h post-UV exposure and is restored within 6 h after UV irradiation (Fig. S3F). RIP assays revealed that PrP3, a U4/U6 snRNP is released from U4 and U6 snRNAs within 2 h post-UV irradiation and re-associates with the snRNAs 6 h post-UV irradiation (Fig. S3G). XAB2 is released from U4, U5, and U6 snRNAs and from all pre-mRNAs tested 2 h post-UV irradiation and its association with UsnRNAs is restored 24 h after UV exposure (Fig. 5a, b). Release from UsnRNAs was also evident for AQR, which persists for 6 h after UV irradiation (Fig. S3H). Importantly, the release of XAB2 from UsnRNAs is not accompanied by its disassociation from other protein components of the spliceosome machinery, i.e., PRP8, PRP19, BCAS2, and SNRP40 (Fig. 5c) and is consistent with the substantial increase in intron retention for several transcripts tested in UV-irradiated cells (Fig. S4A). In contrast to the ATM-dependent UV-induced disassociation of mature spliceosome from nascent transcripts[11], selective inhibition of ATM (ATMi) with KU-55933 inhibitor[41] or of ATR (ATRi) with NU6027 in cells treated with UV did not rescue the release of XAB2 from U4 and U6 snRNAs (Fig. 5d). These findings indicate that the DNA damage-driven release of XAB2 from RNA is not mediated through active DDR signaling. Next, we reasoned that DNA lesions altering the DNA double helix itself and/or interfering with transcription in cis are causal to the aberrant release of XAB2 from UsnRNAs or the pre-mRNAs. To test this, we first treated UV-irradiated cells with BrU; the latter revealed that RNA synthesis recovery is delayed 2 h post-UV irradiation and resumes 6 h—post-UV exposure to levels similar to those seen in untreated wt. controls (Fig. S4B). Next, we made use of MEFs derived from CPD photolyase transgenic mice[42]. Photolyases directly revert photolesions into undamaged bases using visible light energy (i.e.

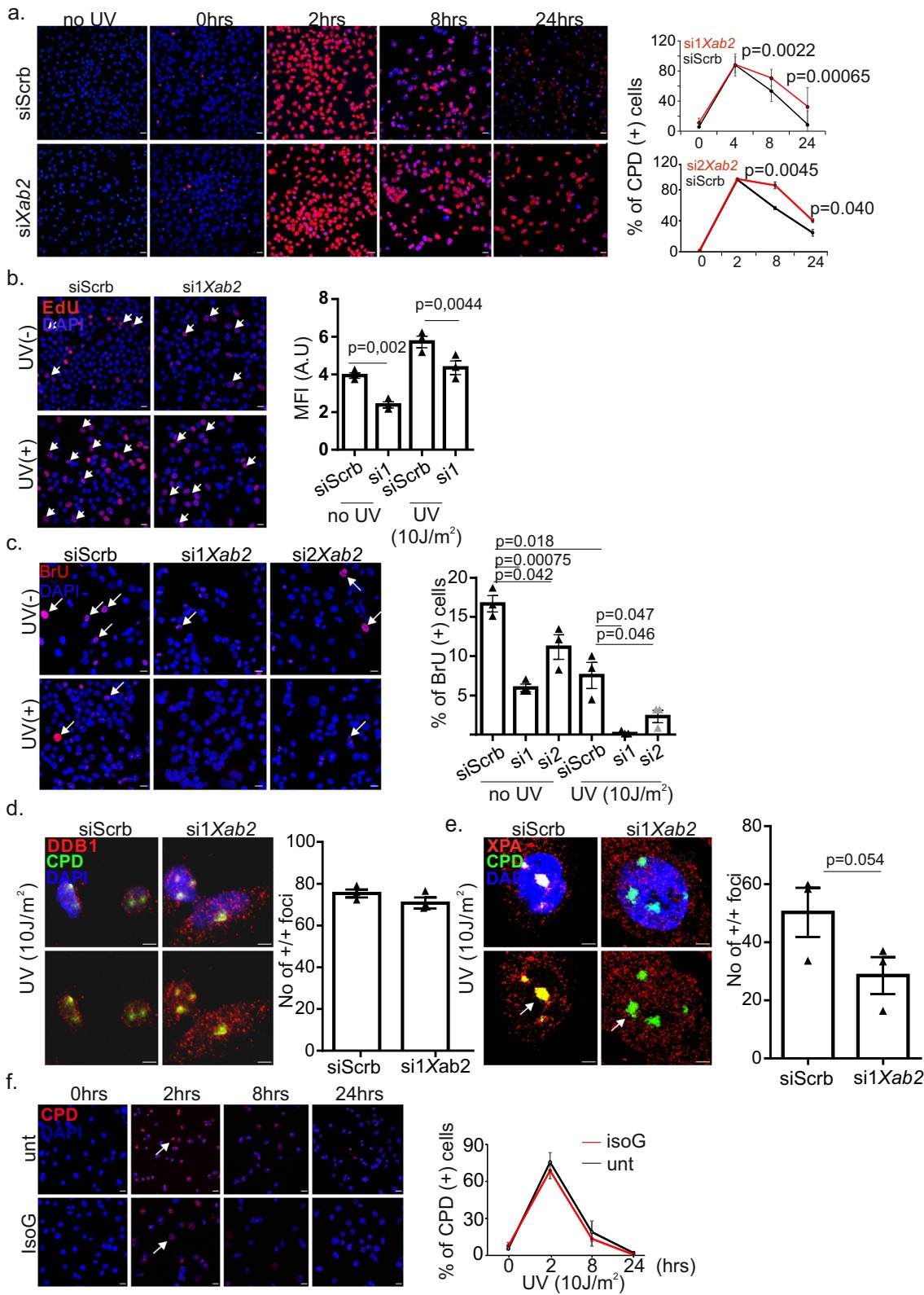

photoreactivation; PR) and display substrate specificity for UV-induced 6-4PPs or CPDs[43]. Using this system, CPD photolyase transgenic MEFs were exposed to UV irradiation and were subsequently kept in the dark (UV-induced CPDs remain in the mammalian genome) or exposed to PR light (UV-induced CPDs are repaired) for 60 min post-UV irradiation (Fig. 5e). Importantly, we find that the exposure of UV-irradiated MEFs to PR light substantially rescues the XAB2 RIP signals in U4 and U6 snRNAs (Fig. 5f) and in pre-mRNAs tested (Fig. 5g) indicating that the presence of DNA helix-distorting lesions inadvertently affects the process of pre-mRNA splicing in the mammalian genome. In line, inhibition of splicing in IsoG-treated cells or of transcription elongation blockage in DRB-treated cells triggers the substantial release of XAB2 from

**Fig. 3 XAB2 is required for NER. a** Immunofluorescence detection and quantification of CPDs in *siScrb* and si1 and si2*Xab2* treated cells at 4, 8, and 24 h post-UV irradiation (10 J/m$^2$) (siScrb(si1), 4 h/8 h/24 h: 1399,1593, 1822 cells respectively, siScr (si2), 4 h/8 h/24 h: 3008, 1550, 4079 cells respectively, si1*Xab2*, 4 h/8 h/24 h: 1531, 1399, 1285 cells respectively, si1*Xab2* 4 h/8 h/24 h: 4950, 590, 4000 cells respectively/3 independent experiments in all cases. Scale bar 15 μm. **b** Immunofluorescence detection of unscheduled DNA synthesis in siScrb ($n = 184$ cells/3 independent experiments) and si*Xab2* ($n = 150$ cells/3 independent experiments) cells treated with 5-ethynyl-2′-deoxyuridine (EdU) at 48 h post-transfection and in siScrb ($n = 134$/3 independent experiments) and si*Xab2* ($n = 166$ cells/3 independent experiments) cells treated with EdU at 48 h post-transfection and 2.5 h post UV irradiation (10 J/m$^2$). Scale bar 15 μm. **c** Immunofluorescence detection of RNA synthesis recovery in siScrb and si*Xab2* cells treated with 5-bromouridine (BrU) at 48 h post-transfection with (siScrb: 1826 cells, si1*Xab2*: 860 cells, si2*Xab2*: 1380 cells/3 independent experiments in all cases) or without UV irradiation (10 J/m$^2$) (siScrb: 1710 cells, si1*Xab2*: 1475cells, si2*Xab2*: 794 cells/3 independent experiments in all cases). Scale bar 15 μm. **d** Immunofluorescence analysis of DDB1 and CPDs in si*Xab2* (113 cells/3 independent experiments) and siScrb (127 cells/3 independent experiments) cells exposed to UV irradiation (10 J/m$^2$). **e** Immunofluorescence detection of XPA and CPDs in siScrb (177cells/3 independent experiments) and si*Xab2* cells (319 cells/3 independent experiments) exposed to UV irradiation (10 J/m$^2$). *p*-value was calculated by one-tailed Student's *t*-test. **f** Immunofluorescence detection and quantification of CPDs in untreated and IsoG treated cells at 4 h (untreated: 2071 cells, IsoG treated: 2198 cells/3 independent experiments in both cases), 8 h (untreated: 1997 cells, IsoG treated: 1837 cells/3 independent experiments in both cases) and 24 h (untreated: 1124 cells, IsoG treated: 1238 cells/3 independent experiments in both cases) post-UV irradiation (10 J/m$^2$). Scale bar 15 μm. All scatter and bar blots in this manuscript are presented as mean ± SEM. *P* values were calculated by two-tailed Student's *t* test, unless otherwise indicated. Scale bar 5 μm, unless otherwise indicated. Error bars indicate SEM among three biological replicates in all cases. Source data provided as a Source Data file.

U4 snRNA (Fig. S4C–D). XAB2 and AQR are also released from U6 snRNA (Fig. S4D) and all pre-mRNAs tested after DRB treatment (Fig. S4E). Similar findings, albeit to a lesser magnitude, were observed upon inhibition of transcription initiation in TPL-treated cells (Fig. S4D–E). Likewise, we find the aberrant release of XAB2 from UsnRNAs in MEFs treated with illudin S known to induce DNA lesions that are ignored by global-genome NER and are exclusively processed by transcription- and replication-coupled repair pathways (Fig. 5h)[44]. The latter prompted us to test whether XAB2 binding to U4 and U6 snRNAs is also altered in *Csb^{m/m}* mice carrying an inborn defect in TC-NER[45]. Using P15 *bXab2*/*Csb^{m/m}* livers, we find that the XAB2 RIP signals on U4 and U6 snRNAs as well as on pre-mRNAs *Cdk7*, *Ercc1* and the highly transcribed *Ghr* and *Igf1* genes are substantially decreased in the P15 livers of *bXab2*/*Csb^{m/m}* animals (Fig. 5i and Fig. S4F).

**Abrogation of XAB2 promotes R-loop formation triggering DNA damage.** R-loops are generated during transcription when nascent RNA exits RNA polymerase and pairs with its complementary DNA template to form a stable RNA–DNA hybrid that displaces single-stranded DNA (ssDNA)[46]. Increased formation of R-loops is frequently observed in cells that are defective in DNA repair or RNA processing[47–49]. A DNA–RNA immunoprecipitation (DRIP) approach followed by treatment with RNase H (RNH; it digests RNA in DNA–RNA hybrids eliminating R-loops) revealed that, upon UV irradiation, R-loops accumulate in *Cdk7* and *Ercc1* genes whose pre-mRNAs were previously shown to associate with XAB2 (Fig. 6a). Inhibition of ATM led to a detectable yet mild decrease in the formation of R-loops indicating that defective DDR signaling alone cannot fully justify the observed accumulation of R-loops in UV-irradiated cells (Fig. 6a). Consistent with our DRIP analysis on UV-irradiated cells (Fig. 6a), confocal microscopy studies revealed that UV irradiation triggers the formation of R-loops in wt. cells that further accumulate when *Xab2* is silenced (Fig. S4G); the latter leads to the substantial increase in the number of γH2Ax (+) 53BP1 (+) cells (Fig. S4H). This and our finding that R-loops accumulate substantially in MEFs treated with IsoG (Fig. 6b, Fig. S4I) prompted us to further explore the functional role of XAB2 in R-loop processing. Similar to UV irradiation or IsoG treatment in MEFs (Fig. 6a, b), we find that knockdown of XAB2 triggers the formation of R-loops in cells; importantly, treatment of cells with RNase H post-fixation led to a substantial decrease in R-loops confirming the validity of these findings (Fig. 6c,

Fig. S5A). Treatment of cells with IsoG also led to the substantial increase of γH2Ax (+) 53BP1 (+) cells (Fig. 6d) as well as to higher γH2Ax protein levels compared to untreated controls (Fig. 6e), indicating that a defect in splicing leads to persistent DNA damage accumulation. Likewise, RNAi-mediated ablation of XAB2 with two distinct siRNAs increased substantially the number of γH2Ax (+) 53BP1 (+) cells (Fig. 6f). As R-loops often lead to persistent DNA breaks, we next tested whether genome instability in si*Xab2* cells is a result of R-loop accumulation. Immunofluorescence studies revealed that protein transfection of RNase H in si*Xab2* cells leads to a decrease in R-loops (Fig. 6g) and in the number of γH2Ax (+) 53BP1 (+) cells (Fig. 6h).

**XAB2 interacts with R-loop processing factors and is recruited to RNA:DNA hybrids under conditions that favor R-loop formation.** Highly transcribed loci are prone to R-loop formation[50,51]. To explore the possibility of a direct association of XAB2 with R-loops and R-loop processing factors, independently of its role in splicing, primary MEFs were treated with all-trans-retinoic acid (tRA), a pleiotropic factor known to activate transcription during cell differentiation and embryonic development[52]. We find that transcription activation in tRA-treated MEFs leads to the substantial accumulation of R-loops in these cells, which becomes further pronounced when cells are treated with tRA and illudin S (Fig. 7a). DRIP analysis with primers spanning the promoter or 3′ end regions of genes (Fig. 7b) where the frequency of RNA–DNA hybrids is expected to be higher revealed the significant accumulation of R-loops in tRA-induced gene targets i.e. *Rarb2*, *Stra6* and in the intronless tRA-induced gene *Fibin*, but not in the tRA non-inducible *Chordc* gene (Fig. 7b, c). Further DRIP analysis revealed that R-loops accumulate in the promoter regions of tRA regulated genes when cells are treated with tRA or with tRA and either illudin S or UV irradiation or DRB (Fig. S5B). Instead, we find that the levels of R-loops in cells decrease after tRA/DRB-treatment or tRA/UV irradiation in the gene bodies of tRA-induced *Rarb2* and *Stra6* genes (Fig. S5C). To confirm that XAB2 and R-loops co-occupy the tRA-responsive *Stra6* gene and the tRA non-inducible *Chordc* gene, we next employed native bioXAB2 pulldown approach followed by S9.6 DRIP in tRA- and tRA/illudin S-treated MEFs. This strategy revealed that bXAB2 is recruited preferentially on the 3′ end region of *Stra6* gene compared to the tRA non-inducible *Chordc* gene in tRA-treated MEFs and that bioXAB2/DRIP signals are higher when cells are treated with illudin S (Fig. 7d). Native ChIP signals prior to S9.6 DRIP revealed that bXAB2 is recruited on the coding region of the tRA-induced

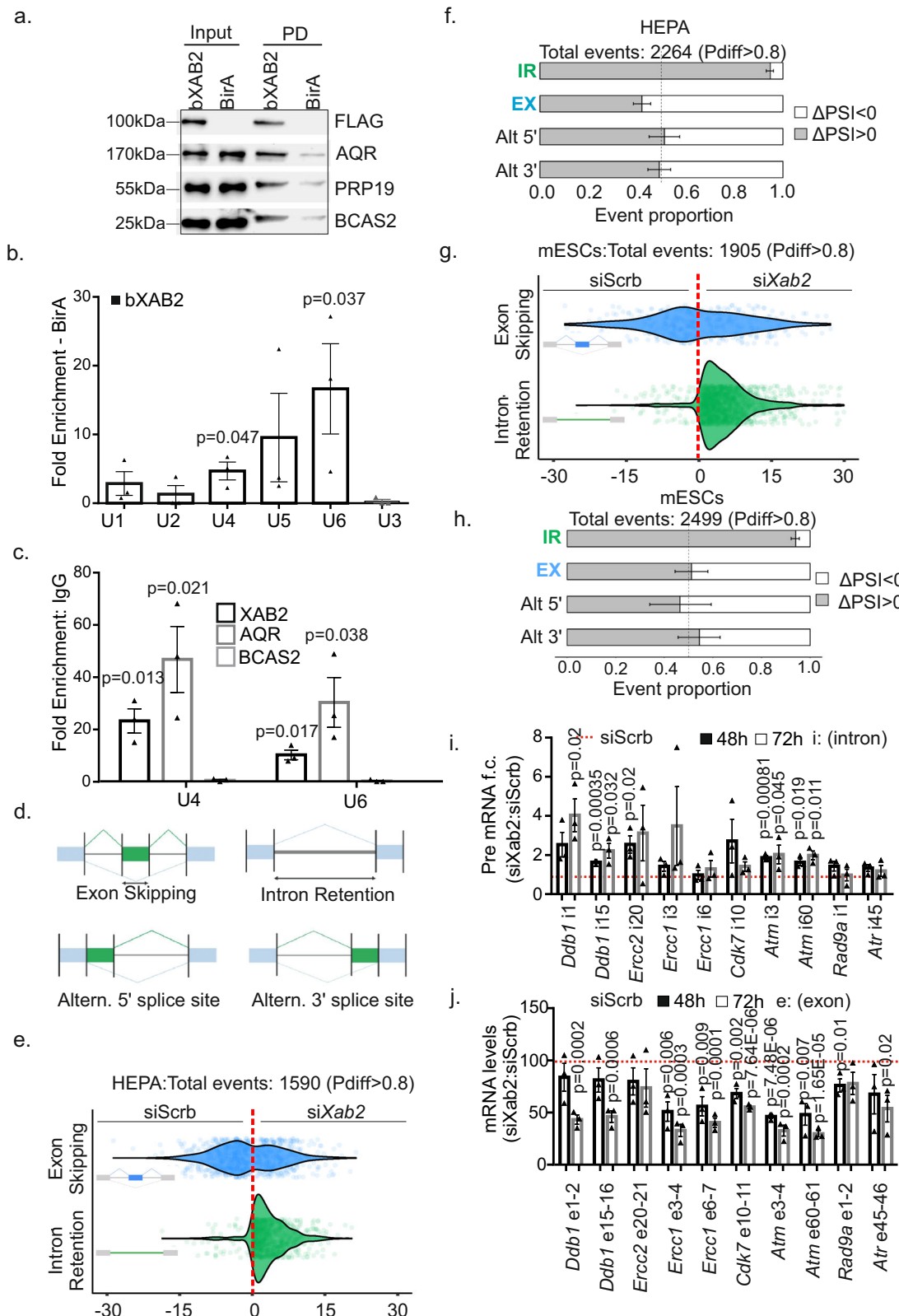

*Stra6* gene but not on the coding region of the tRA non-inducible *Chordc1* gene (Fig. S5D). Similar to the DNA damage-driven release of XAB2 from RNA targets (Fig. 5a, b), XAB2 ChIP signals are substantially reduced when tRA-treated MEFs are also exposed to illudin S accumulating transcription-blocking DNA lesions (Fig. S5D). In agreement with the role of XAB2 in NER (Fig. 3a–e), we find that the association of XAB2 with R-loops is

significantly reduced in the promoter regions of tRA-responsive genes *Stra6* and the intronless *Fibin* and *Sstr4* genes in tRA/UV-treated cells when compared to non-irradiated, tRA-treated control cells (Fig. S5E–F). Instead, the association of XAB2 with R-loops remained unaltered in the tRA non-inducible *Chordc* gene (Fig. S5E). Importantly, the decrease in the association of XAB2 with R-loops on tRA/UV-treated cells was not due to a

**Fig. 4 XAB2 is part of a core spliceosome complex. a** bXAB2 pulldowns and western blot analysis of AQR, FLAG-tagged XAB2 (FLAG), PRP19, and BCAS2 proteins in P15 bXAB2 livers. **b** RNA pulldowns with bXAB2 and BirA (control) on U1-U6 UsnRNAs (as indicated). U6 *p*-value calculated by one-tailed Student's *t*-test. **c** RNA immunoprecipitation with XAB2, Aquarius and BCAS2 on U4 and U6 snRNAs (as indicated). **d** Schematic representation of splicing decisions, i.e., exon skipping, intron retention, and alternate (Altern.) selection of 5′ or 3′ splice sites. **e, g** Graph depicting the differences in Percent Spliced-In (PSI) between si*Xab2* and siScrb HEPA and mESCs cells respectively (ΔPSI > 0: increased alternative exon/intron inclusion in si*Xab2* and ΔPSI < 0: decreased alternative exon/intron inclusion) in si*Xab2* compared to siScrb cells. **f, h** Distribution of the direction of alternative splicing changes between si*Xab2* and siScrb HEPA and mESCs cells, respectively. Represented is the observed proportion of positive (ΔPSI > 0, in gray) values for each event type, with the error bar reflecting its 95% confidence interval, based on a Pearson's chi-squared proportion test as explained in "Methods" (**i**). Pre-mRNA levels and (**j**) mRNA levels of transcripts with retained introns in si*Xab2* and siScrb HEPA cells at 48 and 72 h post-transfection (as indicated). The images shown in Fig. 4a, e, g are representative of experiments that were repeated three times. The RIP signals are shown as fold enrichment (F.E) of % input antibody or bXAB2 over % input control antibody (IgG) or BirA. All scatter and bar blots in this manuscript are presented as mean ± SEM. *P* values were calculated by two-tailed Student's *t* test, unless otherwise indica*t*ed. Error bars indicate SEM among three biological replicates in all cases. Source data provided as a Source Data file.

decrease in the levels of R-loops; previous DRIP analysis on tRA-responsive gene promoters confirmed that R-loops accumulate under the same experimental conditions (Fig. S5B). Further work revealed that XAB2 dissociates from R-loops in tRA-treated bXAB2/*Csb*^*m/m* and bXAB2/*Xpc*^−/− MEFs exposed to UV irradiation (Fig. S5G); in both genotypes, the levels of R-loops were comparable between tRA/UV-treated cells and tRA-treated, non-irradiated controls (Fig. S6A). Lastly, we find that transfection of illudin S-treated MEFs with recombinant RNaseH1 restores the release of XAB2 from UsnRNAs (Fig. S6B).

R-loops are known to be actively processed by the NER structure-specific endonucleases XPF and XPG[53,54]. PRP19 may act as a sensor of RPA-ssDNA moieties that emerge naturally when R-loops are formed during ongoing transcription[21]. Using a series of bXAB2 pulldown under basal transcription conditions or upon tRA treatment in primary MEFs, we find that XAB2 interacts with XPF, XPG, and PRP19 proteins (Fig. 7e, Fig. S6C). Unlike with PRP19, we find that the interaction of XAB2 with XPG or XPF is abolished when cells are treated with RNase H, indicating that the XAB2-XPF-XPG complex requires the presence of R-loops (Fig. 7e, Fig. S6C). The interaction of XAB2 with bXPF and XPG independently of DNA was also evident with bXPF pulldowns in tRA-treated cells albeit to a lesser extent (Fig. S6D). Instead, we find that the interaction of XAB2 with XPF and XPG is lost in tRA-treated cells exposed to UV irradiation (Figs. S6E, S6F). S9.6 DRIP-Western analysis under basal transcription conditions or upon tRA treatment reveals that the XAB2 complex is recruited on R-loops (Fig. 7f, Fig. S6G). A similar analysis reveals that the XAB2 complex is released from R-loops in UV-irradiated MEFs (Fig. 7g) as well as in tRA/UV-treated MEFs (Fig. S6H) indicating the high affinity of XPF and XPG endonucleases for UV-induced DNA lesions. Importantly, neither XPF nor XPG associate with R-loops when XAB2 is silenced (Fig. 7h). Since the recruitment of XPF and XPG on R-loops requires XAB2 and both endonucleases were previously shown to process R-loops into DNA breaks[54,55], it is surprising that *Xab2* knockdown leads to an increase in the number of γH2AX + 53BP1+ cells. Alternatively, these findings could reflect the known role of XAB2 in HR[24] or the possibility that the accumulation of R-loops in si*Xab2* cells could spontaneously lead to DNA breaks due to transcription-DNA replication conflicts in dividing cells[56]. In support, we find that the number of γH2AX + 53BP1+ cells (due to R-loops alone), is reduced in serum-starved MEFs compared to proliferating corresponding control cells (Fig. 7i, Fig. S6I, Fig. S7A). To test whether XAB2 interaction with XPF and XPG on R-loops requires functional NER, we also performed S9.6 DRIP in wt. and totally NER-defective *Xpa*^−/− MEFs (Fig. S7B–C; as indicated) followed by western blotting for XPA (for wt cells), XPF and XPG. We find that XPA does not associate with R-loops and that recruitment of NER

endonucleases on R-loops does not require XPA. Nevertheless, we find that R-loops accumulate in tRA-treated *Xpa*^−/− MEFs compared to wt. controls (Fig. S7D) likely reflecting the contributing role of irreparable DNA lesions (owing to the NER defect in *Xpa*^−/− cells) on R-loop accumulation. In support, we find an enrichment of R-loops on the promoters of highly transcribed *Igf1* and *Ghr* genes in P15 *Csb*^*m/m* livers compared to littermate controls (Fig. S7E). Similar to the reduction of XAB2 ChIP signals in illudin S-treated primary MEFs (Fig. S5D) we find that bXAB2 ChIP signals are substantially reduced on *Ghr* and *Igf1* gene promoters in P15 bXAB2/*Csb*^*m/m* livers compared to age-matched bXAB2 control livers (Fig. S7F). Finally, a sequential native bXAB2 pulldown followed by S9.6 DRIP in P15 bXAB2 and bXAB2/*Csb*^*m/m* livers revealed that XAB2 interacts with R-loops on the promoters of *Igf1* and *Ghr* genes in vivo (Fig. 7j). Thus, XAB2 interacts with XPF and XPG endonucleases outside NER and is recruited to RNA:DNA hybrids under conditions that promote R-loop formation (Fig. 7k).

## Discussion

Inborn defects in RNA processing or DNA repair associate with progeria[27,28,57], metabolic defects[58–63], neurodegeneration[64–67], and cancer[68–72], arguing for functional links between genome maintenance and the splicing machinery in development or disease. Using an in vivo biotinylation tagging approach in mice, we find that XAB2 interacts with components of the core spliceosome as well as with DDB1, involved in UV-induced DNA damage recognition and XPA that plays a prominent role in the assembly of the NER incision complex at sites of DNA damage[67,73]. As DDB1 is successfully recruited to UV-induced CPDs in si*Xab2* cells, the implication of XAB2:DDB1 interaction remains elusive. However, in si*Xab2* cells, XPA fails to form visible foci and to be recruited to UV-induced DNA lesions. In support of a defect in NER, si*Xab2* cells have decreased UDS and delayed RNA synthesis recovery following UV-induced DNA damage. Consistently, UV-induced CPDs remain unrepaired even 24 h post-UV irradiation in si*Xab2* cells. Arguably, the defect in splicing does not interfere with the DNA repair defect in si*Xab2* cells because exposure of cells to a pre-mRNA splicing inhibitor has no detectable impact on the repair kinetics of UV-induced CPDs. We find that XAB2 is recruited to UsnRNAs and pre-mRNAs and that it is released from all RNA targets tested when cells are treated with UV-induced DNA damage or with the genotoxin illudin S. Intriguingly, the release of XAB2 from RNA following DNA damage is not rescued when ATM or ATR are inhibited but only when UV-induced CPDs are efficiently removed through PR-dependent repair. The latter supports previous observations showing that DNA damaging agents affect the subcellular localization of splicing factors or their association with

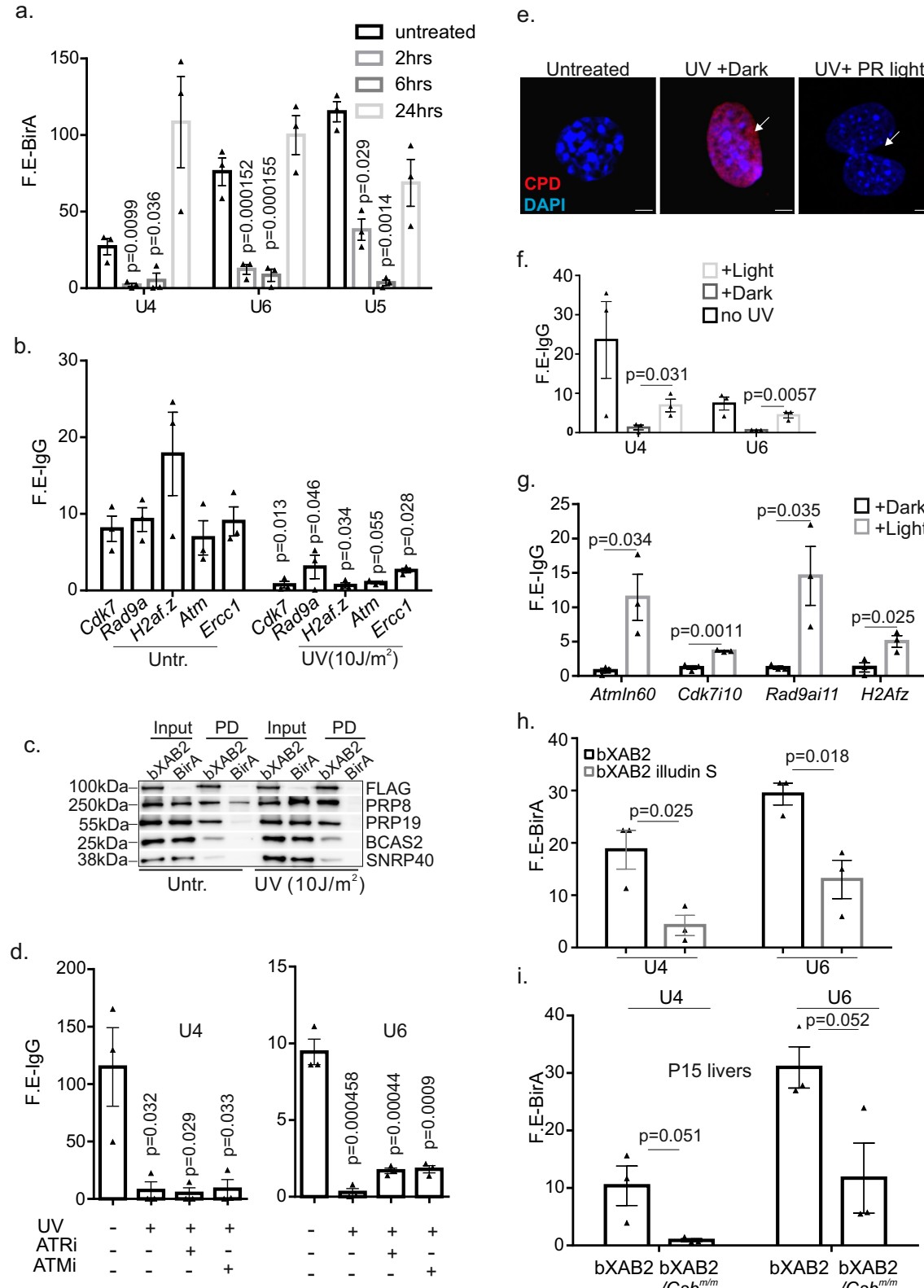

nascent transcripts[74–77] and that ATM (and likely ATR) function downstream of the spliceosome displacement[11].

Nascent pre-mRNAs that associate with transcription complexes blocked at DNA lesions are released from late-spliceosome components and hybridize to the template strand of the melted DNA to form RNA:DNA duplexes (R-loops), a structure that slows transcription and activates DDR[77]. Increased formation of

R-loops is frequently observed in cells that are defective in DNA repair or RNA processing[47–49] leading to DNA breaks[54], transcription stalling, and DDR activation[77]. In support, we find that R-loops accumulate in primary MEFs following DNA damage or upon treatment with a pre-mRNA splicing inhibitor. Knockdown of *Xab2* leads to the substantial increase in R-loop formation and to an R-loop-dependent accumulation of γH2AX and 53BP1 in

**Fig. 5 XAB2 is released from U snRNPs and pre-mRNAs upon DNA damage. a** RNA immunoprecipitation of XAB2 on U4, U5, and U6 snRNAs at 2, 6, and 24 h post-UV irradiation. **b** RNA immunoprecipitation of XAB2 on pre-mRNAs in untreated and UV-irradiated HEPA cells (as indicated). **c** bXAB2 pulldowns followed by western blot analysis with FLAG (for bXAB2), PRP8, PRP19, BCAS2, and SNRP40 in UV-irradiated (10 J/m$^2$) and control MEFs. The images shown on Fig. 5c are representative of experiments that were repeated three times. **d** RNA immunoprecipitation of XAB2 on UsnRNAs in untreated, UV-irradiated cells and UV-irradiated cells pre-treated with inhibitors against ATM (ATMi) and ATR (ATRi; as indicated). **e** Immunofluorescence detection of UV-induced CPDs in UV-irradiated photolyase transgenic MEFs exposed to 1 h of PR light (CPDs are repaired) or dark (CPDs remain) as indicated. The images shown are representative of experiments that were repeated three times. **f** RNA immunoprecipitation (RIP) of XAB2 on U4 and U6 snRNAs in UV-irradiated photolyase transgenic MEFs exposed to 1 hour of PR light (CPDs are repaired) or dark (CPDs remain); **g** RNA immunoprecipitation (RIP) of XAB2 on pre-mRNAs in UV-irradiated photolyase transgenic MEFs exposed to 1 h of PR light (CPDs are repaired) or dark (CPDs remain); **h** bXAB2 RNA pull downs on U4 and U6 snRNAs in untreated or illudin S-treated BirA and bXAB2 MEFs; **i** bXAB2 RNA pull downs on U4 and U6 snRNAs in P15 bXAB2 or bXAB2;*Csb*$^{m/m}$ mouse livers. RIP signals are shown as fold enrichment (F.E) of % input of antibody or bXAB2 over % input of control antibody (IgG) or BirA. All scatter and bar blots in this manuscript are presented as mean ± SEM. *P* values were calculated by two-tailed Student's *t* test, unless otherwise indicated. Error bars indicate SEM among three biological replicates in all cases. Source data provided as a Source Data file.

cells. Indeed, the number of γH2AX (+) 53BP1 (+) cells is significantly reduced when si*Xab2* cells are protein transfected with RNaseH1 known to eliminate R-loops. Using a series of IP strategies, we find that XAB2 interacts with XPG and XPF endonucleases and that the XAB2 is required for the trimeric protein complex to be recruited on RNA:DNA hybrids under conditions that favor R-loop formation. The latter supports previous observations indicating that the NER endonucleases XPF and XPG actively process unscheduled R-loops into DNA breaks[54,78]. The interaction of XAB2 with XPG or XPF as well as the recruitment of the XAB2 complex on RNA:DNA hybrids are also observed in NER-defective *Xpa*$^{-/-}$ MEFs, indicating that R-loop processing does not require functional NER. Similar to RNA, transcription-blocking DNA lesions trigger the release of XAB2 from DNA sites. However, we find that bXAB2 ChIP signals followed by DRIP are substantially higher on the 3′ end region of the tRA-inducible *Stra6* gene and increase even further when cells are also treated with illudin S. Likewise, DRIP signals followed by bXAB2 ChIP are higher on the R-loop-enriched promoter regions of *Igf1* and *Ghr* genes in P15 *Csb*$^{m/m}$ livers indicating that XAB2 preferentially binds on DNA sites that associate with RNA. Taken together, our findings suggest that the splicing factor XAB2 is released from UsnRNAs and pre-mRNAs and together with XPG and XPF, it is recruited on RNA:DNA hybrids under conditions that promote R-loop formation, e.g., transcription induction or DNA damage (Fig. 7k). The finding that XAB2 is released from RNA targets in progeroid *Csb*$^{m/m}$ developing livers makes it attractive to test whether DNA damage-driven changes in RNA processing are causal to the premature onset of age-related pathological symptoms seen in TC-NER progeroid syndromes and during natural aging.

## Methods

**Generation of biotin-tagged XAB2 animals.** To generate the targeting vector for the insertion/knock-in of the Avi tag cassette before the stop codon of the last exon of the XAB2 gene for the generation of the avXAB2 knock-in mice, PCR products were first amplified using Phusion High- Fidelity DNA Polymerase (NEB). The avi-tag was sub cloned in pBSSK (EcoRI/Hind III 0.18 kb). A triple ligation reaction was set up using the fragments: 5′ homology sub cloned in two fragments (XbaI/BamHI 2.4 kb and BamHI/ExoRI 1.2 kb); avi tag (EcoRI/Hind III 0.18 kb); pBSSK-avi tag (XbaI/ExoRI 2.9 kb). The lox-neomycin-lox cassette (HindIII/SalI 1.5 kb) was subsequently cloned into the vector followed by cloning of the 3′ homology region (SalI fragment 3.3 kb). Finally, the MC1-TK gene (SacII 1.8 kb) was inserted into the vector for negative selection. The final targeting vector was linearized using NotI and used for embryonic stem cell electroporation. 129/SV embryonic stem cells carrying the Protamine 1-Cre transgene were maintained in their undifferentiated state (LIF-ESGRO 10$^7$ units) and grown on a feeder layer of gamma-irradiated (3500 rads) G418$^r$ primary MEFs. Electroporation (400 V, 25 μF) of $0.8 \times 10^7$ embryonic stem cells with 50 μg of NotI linearized targeting vector (2 mg mL$^{-1}$) was performed and homologous recombined clones were selected with G418 (300 μg mL$^{-1}$) and ganciclovir (2 μM). G418-resistant embryonic stem cell clones were subjected to Southern blot analysis and hybridized with 5′ and 3′ probes from their homology region. Genomic DNA from embryonic stem cell clones was digested overnight with EcoRV (MINOTECH Biotechnology) and

resolved on 1% agarose gels. Samples were immobilized on Hybond-NC nylon membranes (Amersham Bioscience) and hybridized with probes with [$^{32}$P] dCTP (Izotop). 5′ (1.1 kb NcoI/EcoRI) and 3′ (1.2-kb BglII/Hind III) specific probes flanking the last exon of the Xab2 gene were used to identify the targeted (6.4 kb or 7.5 kb) and wild-type allele (15.8 kb). Clones with the correct homologous recombination were expanded to confirm their integrity and karyotyped to verify their euploid karyotype. Positive clones tested negative for mycoplasma (Venor GeM) were used for C57/BL6 blastocyst injection to generate chimeric mice. Chimeric males were bred to C57BL/6 wild-type females for germline transmission. Offspring were screened by PCR for neo-deletion using primers F1 and R1 (Fig. 1a, supplementary Table 1). Expression of Protamine-1 Cre transgene in the male germline resulted in the deletion of the floxed neomycin gene in all the first pups born, leaving behind a single loxP site after the avi tag cassette. The Cre recombinase transgene, derived from the PC3 embryonic stem cell background, was bred out in the process of backcrossing to C57BL/6 mice. Biotin-tag XAB2 knock-in mice were further crossed to transgenic BirA transgenic mice[31]. Mice were kept on a regular diet and housed at the IMBB animal house, which operates in compliance with the 'Animal Welfare Act' of the Greek government, using the 'Guide for the Care and Use of Laboratory Animals' as its standard. As required by Greek law, formal permission to generate and use genetically modified animals was obtained from the responsible local and national authorities. The independent Animal Ethical Committee at FORTH approved all animal studies.

**IP assays.** Nuclear protein extracts from 15-day-old livers or cells were prepared as previously described[31] using the high-salt extraction method (10 mM HEPES-KOH pH 7.9, 380 mM KCl, 3 mM MgCl$_2$, 0.2 mM EDTA, 20% glycerol, and protease inhibitors). For IP assays, nuclear lysates were diluted threefold by adding ice-cold HENG buffer (10 mM HEPES-KOH pH 7.9, 1.5 mM MgCl2, 0.25 mM EDTA, 20% glycerol) and precipitated with antibodies overnight at 4 ºC followed by incubation for 2 h with protein G Sepharose beads (Millipore). Mouse or rabbit IgG (Santa Cruz) was used as a negative control. Immunoprecipitates were washed five times (10 mM HEPES-KOH pH7.9, 300 mM KCl, 0.3% NP40, 1.5 mM MgCl2, 0.25 mM EDTA, 20% glycerol and protease inhibitors), eluted and resolved on 10% SDS-PAGE. For bXAB2 and bXPF pulldowns, 0.6–0.7 mg of nuclear extracts were incubated with M-280 paramagnetic streptavidin beads (Invitrogen) as previously described[31].

**Mass spectrometry (MS) studies.** For MS studies, nuclear protein extracts derived from livers of 15-day-old BirA or bXAB2 mice ($n = 3$) were prepared using the high-salt extraction method (10 mM HEPES-KOH pH 7.9, 380 mM KCl, 3 mM MgCl$_2$, 0.2 mM EDTA, 20% glycerol and protease inhibitors). Nuclear extracts derived from BirA animals were used as negative controls. Nuclear lysates were diluted threefold with HENG buffer and digested O/N with benzonase. For pull-down assays used in MS, 0.6–0.7 mg of nuclear extracts were incubated with 40 μl M-280 paramagnetic streptavidin beads (Invitrogen); streptavidin interacts with the biotin moiety that is added by the BirA ligase on the short 15aa biotinylation Avitag of XAB2. Proteins eluted from the beads were separated by SDS/PAGE electrophoresis on an 10% polyacrylamide gel and stained with Colloidal blue silver (ThermoFisher Scientific, USA[79]); SDS-PAGE gel lanes were cut into 2-mm slices. Gel areas that were very dense in protein content or contained a significant amount of a particular protein were cut in smaller slices. Gel slices were subjected to in-gel reduction with dithiothreitol, alkylation with iodoacetamide and digested with trypsin (sequencing grade; Promega)[80,81]; the 'in gel' strategy allows the possibility to verify the molecular weight of the identified proteins, thereby minimizing any false positive hits. Nanoflow liquid chromatography-tandem mass spectrometry (nLC-MS/MS) was performed on an EASY-nLC coupled to an Orbitrap Fusion Tribid mass spectrometer (Thermo) operating in positive mode. Peptides were separated on a ReproSil-C18 reversed-phase column (Dr. Maisch; 15 cm × 50 μm) using a linear gradient of 0–80% acetonitrile (in 0.1% formic acid) during 90 min at a rate of 200 nl/min. The elution was sprayed into the electrospray ionization (ESI)

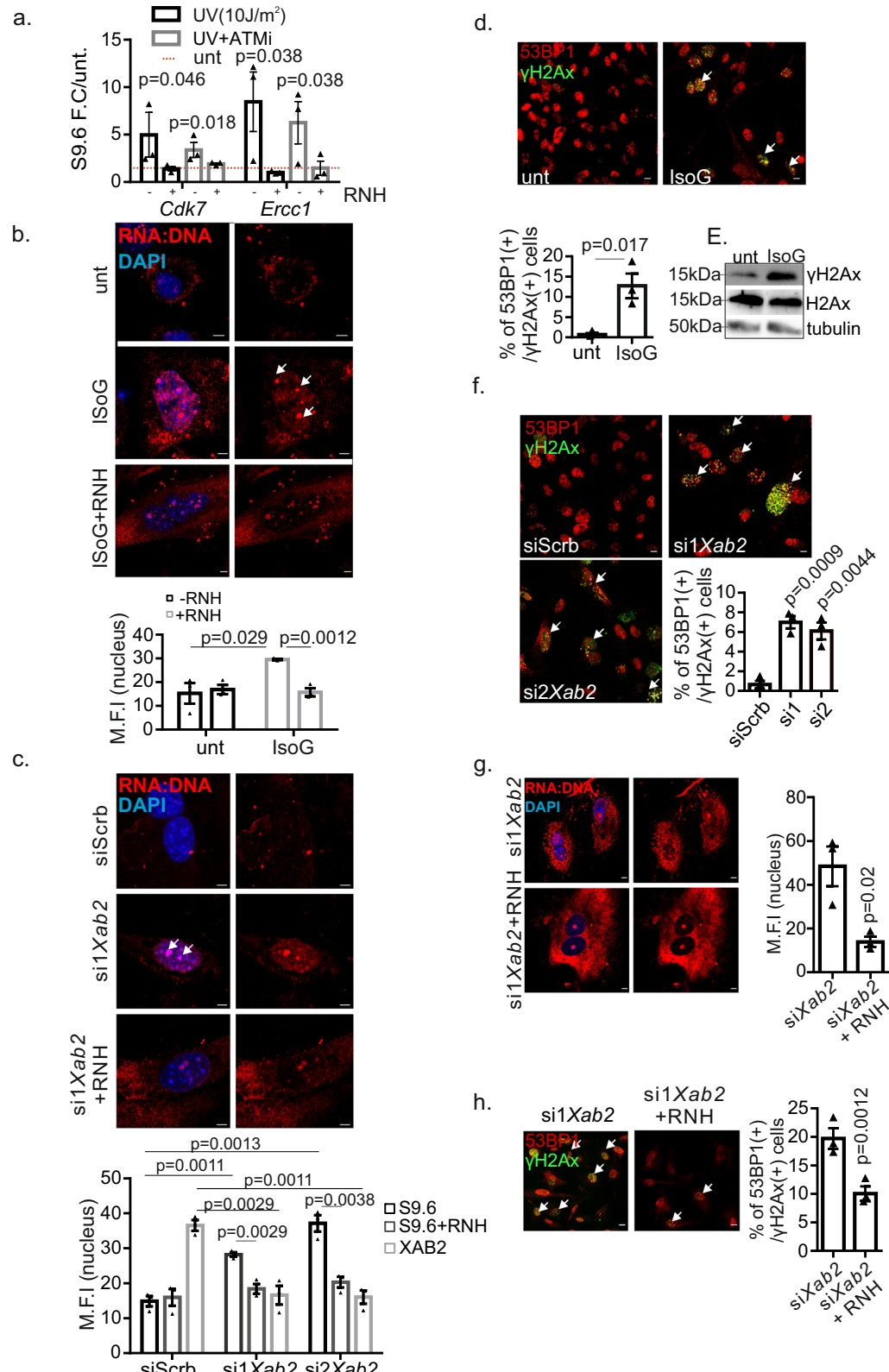

source of the mass spectrometer. Spectra were acquired in continuum mode; fragmentation of the peptides was performed in data-dependent mode by HCD.

**LC-MS/MS data analysis**. Raw mass spectrometry data were analyzed with the MaxQuant software suite[82] (version 1.6.0.16)[81] with the additional options 'LFQ' and 'iBAQ' selected. A false discovery rate of 0.01 for proteins and peptides and a minimum peptide length of seven amino acids were used to discern the significant data obtained from bXAB2 and BirA mouse livers. The Andromeda search engine was used to search the MS/MS spectra against the Uniprot database (taxonomy: *Mus musculus*, release September 2017), concatenated with the reversed versions of all sequences. A maximum of two missed cleavages was allowed. The peptide tolerance was set to 10 ppm and the fragment ion tolerance was set to 0.6 Da for HCD spectra. The enzyme specificity was set to trypsin and cysteine

**Fig. 6 Abrogation of XAB2 leads to R-loop-dependent genomic instability. a** DRIP analysis on promoter regions of *Cdk7* and *Ercc1* genes in untreated and UV-irradiated MEFs with or without RNase H1 (w/o RNH) and pretreated with ATM inhibitor (ATMi), as indicated. *P* values were calculated by one-tailed Student's *t* test. DRIP signals are shown, as fold change over untreated. **b–c** Representative immunofluorescence images and quantification of RNA:DNA hybrids (indicated by the white arrows) in untreated and IsoG (IsoG)-treated wt. MEFs (see supplemental Fig. S4I for lower magnifications) and in siScrb and si*Xab2* MEFs, respectively (see supplemental Fig. S5A for lower magnifications and representative images for the efficiency of *Xab2* silencing in si2*Xab2* cells), w/o RNH. The graph depicts the mean S9.6 fluorescence intensity per nucleus (untreated ∓RNH: *n* = 1478/685 cells, IsoG treated ∓RNH: *n* = 1000/671cells, siScrb ∓ RNH: *n* = 1078/700 cells, si1*Xab2* ∓RNH: *n* = 540/495 cells, si2*Xab2* ∓ RNH: *n* = 640/580 cells/3 independent experiments)). Scale bars, 5 μm. **d–f** Immunofluorescence detection of γH2AX and 53BP1 (white arrowheads) in untreated and IsoG-treated wt. MEFs and in siScrb and si*Xab2* MEFs (see supplemental Fig. S5A for the efficiency of *Xab2* silencing), respectively. The graph represents the percentage of γH2AX+ 53BP1+ cells (untreated: *n* = 1478 cells, IsoG-treated: *n* = 1045 cells, siScrb: *n* = 1095 cells, si1*Xab2*: *n* = 567 cells, si2*Xab2*: *n* = 723 cells/3 independent experiments)). Scale bar, 15 μm. **e** Western blot analysis of phosphorylated and total H2AX levels in untreated and IsoG-treated MEFs. The image shown is representative of experiments repeated three times. **g** Immunofluorescence detection of RNA:DNA hybrids in si*Xab2* MEFs with or without RNaseH1 (RNH) protein transfection. The graph represents the S9.6 fluorescence intensity per nucleus (si*Xab2*∓RNH: *n* = 653/587 cells/3 independent experiments. Scale bar, 5 μm. **h** Immunofluorescence detection of γH2AX and 53BP1 (white arrowheads) in si*Xab2* MEFs with or without RNaseH1 (RNH) protein transfection. The graph represents the percentage of γH2AX+ 53BP1+ cells. (si*Xab2* ∓ RNH: *n* = 525/578 cells/3 independent experiments.).Scale bar, 15 μm. All scatter and bar blots are presented as mean ± SEM. *P* values were calculated by two-tailed Student's *t* test, unless otherwise indicated. Error bars indicate SEM among three biological replicates. Source data provided as a Source Data file.

carbamidomethylation was set as a fixed modification. In case the identified peptides of two proteins were the same or the identified peptides of one protein included all peptides of another protein, these proteins were combined by MaxQuant and reported as one protein group. The generated 'proteingroups.txt' table was filtered for contaminants and reverse hits. For bXAB2 interactor identification, a *t*-test-based statistics was applied on LFQ of proteome derived from bXAB2 and BirA livers. First, the logarithm (log 2) of the LFQ values were taken, resulting in a Gaussian distribution of the data. This allowed imputation of missing values by a normal distribution (width = 0.3, shift = 1.8), assuming these proteins were close to the detection limit. Statistical outliers for the XAB2 samples compared to BirA control were then determined using a two-tailed *t*-test. Multiple testing correction was applied by using a permutation-based false discovery rate (FDR) method in Perseus.

**Cells, colony formation, and unscheduled DNA synthesis assays**. Mouse embryonic stem cells JM8A3N.1 were cultured on gelatinized tissue culture dishes in a medium containing Dulbecco's modified Eagle's medium (DMEM) supplemented with 15% fetal bovine serum (FBS), 50 μg mL⁻¹ streptomycin, 50 U mL⁻¹ penicillin (Gibco), 2 mM L-glutamine (Gibco), 1% non-essential amino acids (Gibco), 0.1 mM β-Mercaptoethanol (Applichem), 1 μM MEK inhibitor PD0325901 (Selleck), 3 μM GSK inhibitor CHIR99021 (Selleck). Primary MEFs (P4) and HEPA cells were cultured in a standard medium containing DMEM supplemented with 10% FBS, 50 μg mL⁻¹ streptomycin, 50 U mL⁻¹ penicillin (Gibco), 2 mM L-glutamine (Gibco). Serum-starved MEFs were grown in DMEM supplemented with 1% FBS for 24 h before treatments. To knockdown *Xab2* an oligo RNA was designed at the position 1685 (si1) and at position 1460 (si2) of the cDNA (Invitrogen). Mouse embryonic stem cells were transfected in suspension with Lipofectamine 2000 (Invitrogen) and subsequently plated in a 60 mm plate. HEPA cells were transfected with Polyplus Jet prime according to the manufacturer's protocol. As a non-targeting control, AllStars negative (Qiagen) was used. MEFs were transfected with Amaxa Mouse Embryonic Fibroblast Nucleofector Kit. Both siRNAs were used at a final concentration of 50 nM. MEFs were protein transfected with recombinant RNase H (NEB) using Project Reagent Transfection Kit (Thermo, #89850). 5units of RNase H were used per 24-well plate. Cells were transfected 1 h before tRA/tRA illudin treatments. Cells were rinsed with PBS, exposed to UVC irradiation at the indicated doses, MMC (10 μg mL⁻¹) (AppliChem), tRA (10 μM) (Sigma-Aldrich), illudin S (50 ngr/ml), IsoG (30–60 μM) and cultured at 37 ºC for 4–12 h prior to subsequent experiments. Pre-incubation with IsoG (30–60 μM), ATM inhibitor (10 μM) and ATR inhibitor (10 μM), started 6 h (IsoG) or 1 h (ATM, ATR inhibitors) before UVC irradiation and lasted throughout the experiment. For cell survival experiments, a total of 200–300 HEPA cells 24 h post knock-down or primary MEFs or were seeded in 10 cm Petri dishes. The next day, MEFs were exposed to MMC treatment for 4 h or to UVC irradiation and incubated for 10 days. Colonies were stained with Coomassie blue (0.2% Coomassie blue, 50% methanol, 7% acetic acid), and the number of colonies was counted and expressed as a percentage of the treated cells relative to that of the untreated control. Three dishes per dose were used and at least three independent survival experiments were performed. For viability experiments, Trypan blue inclusion was used at selected time points post transfection. Culture medium was collected and centrifuged to precipitate dead cells. HEPA cells were detached from the tissue culture dish by trypsin-0.5% EDTA, resuspended in culture medium and merged with the fraction of cells from the medium. The cell suspension was subsequently diluted 1:5 with 0.4% Trypan blue and the number of viable and dead cells was counted in a Neubauer haemocytometer under the microscope. The number of viable cells was expressed as the percentage of viable cells relative to that of the control cells. At least three independent viability experiments per time point

were performed. DNA repair synthesis was determined by 5-ethynyl-2′-deoxyuridine (EdU) incorporation. Primary MEFs grown on coverslips were globally UVC irradiated and incubated for 2.5 h in medium supplemented with 10 mM EdU. After EdU incorporation, cells were washed with PBS followed by fixation with 2% formaldehyde in PBS. Coverslips were blocked for 30 min with 10% FBS in PBS, followed by 1 h incubation with 10 mM sodium ascorbate and 4 mM CuSO₄ containing Alexa Fluor 594 azide (ThermoFischer Scientific A10270) and DAPI staining. The number of EdU-positive cells among at least 200 cells was counted, and the percentage of EdU-positive cells relative to the total number of cells was calculated. DNA transcription sites were labeled as follows. MEFs were grown on coverslips. After treatments cells were washed with ice-cold TBS buffer (10 mM Tris-HCl, 150 mM NaCl, 5 mM MgCl₂) and further washed with glycerol buffer (20 mM Tris-HCl, 25% glycerol, 5 mM MgCl₂, 0,5 mM EGTA) for 10 min on ice. Washed cells were permeabilised with 0,5% TritonX-100 in glycerol buffer (with 25 U/ml RNase inhibitor) on ice for 3 min and immediately incubated at RT for 30 min with nucleic acid synthesis buffer (50mMTris-HCl pH7.4, 10 mM MgCl₂, 150 mM NaCl, 25%glycerol, 25 U/ml RNase inhibitor, protease inhibitors, supplemented with 0,5 mM ATP,CTP,GTP and 0.2 mM BrUTP. After incorporation, cells were fixed with 4% formaldehyde in PBS on ice for 10 min. Immunofluorescence with a-BrdU antibody was performed as described below.

**Immunofluorescence, antibodies, westerns blots, and FACS**. Immunofluorescence experiments were performed as previously described[31,60]. Briefly, cells were fixed in 4% formaldehyde, permeabilized with 0.5% Triton-X and blocked with 1% BSA. After 1-h incubation with primary antibodies, secondary fluorescent antibodies were added and DAPI was used for nuclear counterstaining. Samples were imaged with SP8 confocal microscope (Leica). For local DNA damage infliction, cells were UV-irradiated (60 J m⁻²) through isopore polycarbonate membranes containing 3-μm-diameter pores (Millipore). For CPD immunodetection, nuclear DNA was denatured with 1 M HCl for 30 min. For S9.6 immunofluorescence cells were fixed with ice-cold methanol at −20 °C for 10 min RNase H treatment was performed post-fixation at 37 °C in PBS supplemented with 10–15 Units RNaseH for 30 min. For whole-cell extract preparations, cell pellets were resuspended in 150 mM NaCl, 50 mM Tris pH = 7.5, 5% glycerol, 1% NP-40, 1 mM MgCl and incubated on ice for 30 min. For cell cycle analysis cells were fixed with 70% ethanol for 30 min, washed with PBS, RNase A treated (1 mg/ml) at 37 °C for 30 min and stained with propidium iodide (RT,20 mg/ml) for 1 h. Antibodies against XAB2 (wb: 1:2000, IF: 1:1000), PRP19 (wb: 1:1000, IF: 1:500), β-TUB (wb: 1:2000) were from Abcam. BCAS2 (wb: 1:5000, IF: 1:1000), DDB1 (IF: 1:500) were from Novus. HA (Y-11, western blotting (wb): 1:500), ERCC1 (D-10, wb: 1:500), RAD9A (wb: 1:300), POLII (wb: 1:500, IF: 50), XPA (wb: 1:500), XPF (F-11, wb: 1:500) and XPG (sc12558, wb: 1:200) were from Santa Cruz Biotechnology. γH2Ax (05-636, IF: 1:12.000) and S9.6 (MABE1095, IF: 1:300) was from Millipore. DDB1 (wb: 1:5000), XPC (wb: 1:1000) and CSB (wb: 1:1000) were from Bethyl Laboratories. stp-HRP (wb: 1:12,000) was from Upstate Biotechnology. FLAGM2 (F3165, wb 1:2000) was from Sigma-Aldrich. CPD (IF: 1:50) was Cosmo Bio Ltd (TDM2). BrdU (1:300, IF) was from BD Pharmingen.

**Differential AS analysis**. Global quality of FASTQ files with raw RNA-seq reads was analyzed using *fastqc* v.0.11.5 (https://www.bioinformatics.babraham.ac.uk/projects/fastqc/). *Vast-tools*[83,84] aligning and read processing software was used for quantification of alternative sequence inclusion levels from FASTQ files using VASTD-DB annotation[85] for mouse genome assembly mm9. To ensure sufficient read coverage for AS quantification, we considered only *vast-tools*' events with a minimum mapability-corrected read coverage score of "VLOW" across all samples.

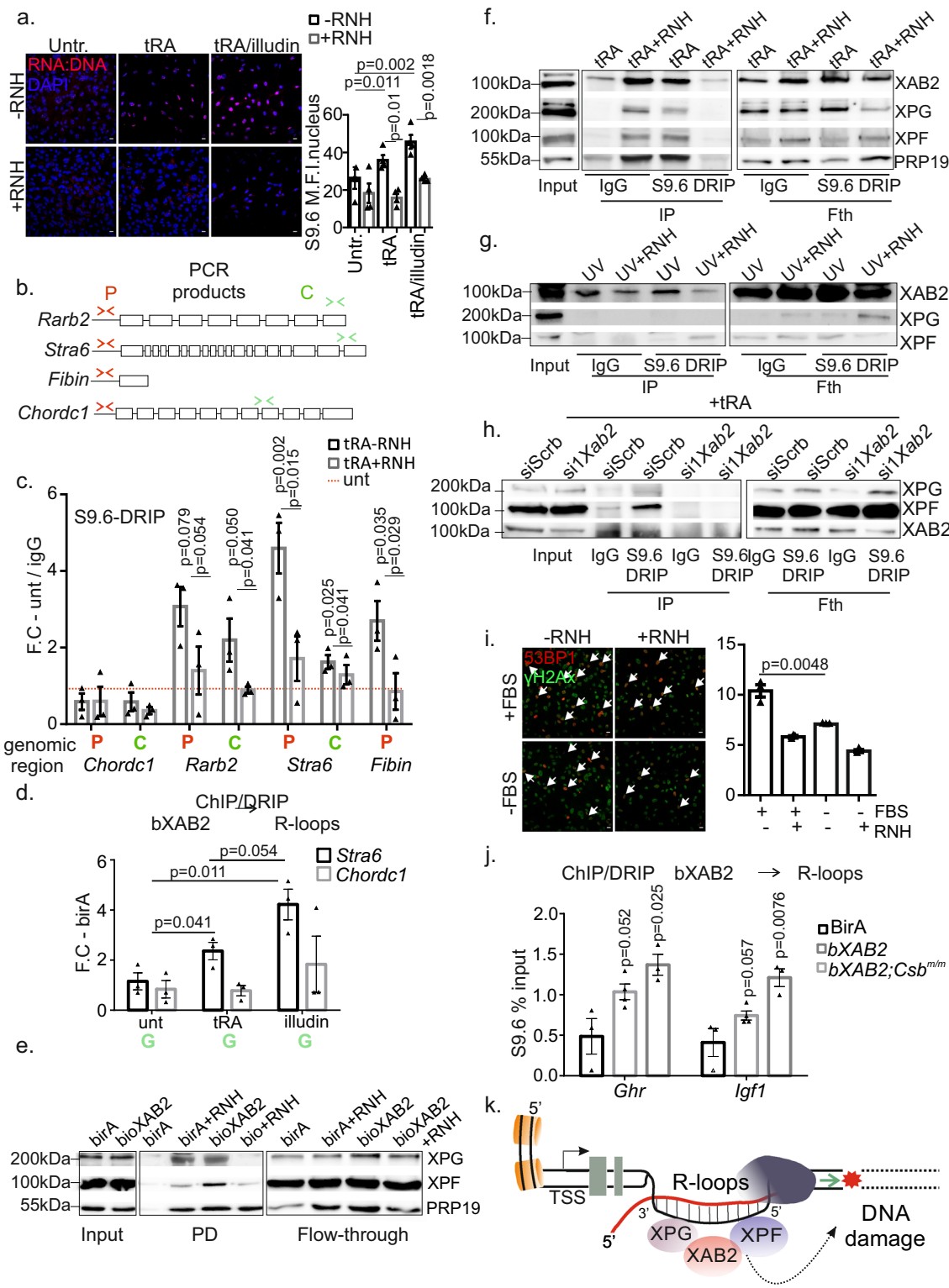

The relative inclusion of an alternative sequence (exon or intron), henceforth called AS event, is based on the number of junction reads supporting inclusion (#inc) and exclusion (#exc), used to quantify percent spliced-in (PSI)[86] values. The beta distribution (conjugate prior probability distribution for the binomial), constrained in the] 0,1 [interval and characterized by a mean value of $\alpha/(\alpha + \beta)$ from the distribution's shape parameters $\alpha$ and $\beta$, was exploited in modeling the precision of each PSI from its supporting coverage (#inc and #exc). We used R function $rbeta$ to emit, for each sample, 500 values from a beta distribution with $\alpha$ = #inc + 1 and $\beta$ = #exc + 1, where 1 is added to ensure that both the shape parameters, $\alpha$ and $\beta$, are different from zero. Since beta distributions get narrower with increasing shape

parameters (for the same PSI value), this parametrization allows the scattering of emitted values to serve as a surrogate for that PSI's dispersion (given the original read coverage supporting it), while the distribution's mean value, (#inc + 1)/(#inc + 1 + #exc + 1), is an approximation of the empirical PSI. For each event, $rbeta$-emitted values were grouped per condition (Control and Xab2 siRNA) and the median of all emitted values per group was used to determine global PSIs per condition and the difference between these, $\Delta PSI = PSI_{Xab2\ siRNA} - PSI_{Control}$. For each AS event, the beta distribution's emitted values were used in calculating the significance of its $\Delta PSI$. First, the difference between randomly ordered Xab2 siRNA and Control vectors of emitted values was calculated. The significance

**Fig. 7 High transcription promotes XAB2 interaction with R-loops. a** Immunofluorescence detection of R loops in untreated, tRA-treated (4 h) and illudin S-treated (1.5 h) MEFs (untreated ∓RNH: $n = 975/705$ cells, tRA treated ∓RNH: $n = 1488/750$ cells, illudin ∓RNH: $n = 1500/730$ cells/4 independent experiments.). **b–c** DRIP analysis of tRA-responsive (*Rarb2*, *Fibin*, and *Stra6*) and non-responsive (*ChordC*) gene promoters (P) or coding regions (**c**) w/o RNHin untreated and tRA-treated MEFs. *p* values between (−) and (+) RNH treated were calculated with one-tailed Student's *t* test. **d** Native ChIP followed by DRIP on the tRA-induced *Stra6* and the tRA-non induced *Chordc1* coding regions in tRA and illudin S/tRA-treated BirA and bXAB2 MEFs. **e** bXAB2 pulldowns (PD) and western blotting for XPF, XPG, and PRP19 in tRA-treated bXAB2 and BirA MEF nuclear extracts w/o RNH. **f–h** S9.6 immunoprecipitation followed by WB for XAB2, XPG, XPF and PRP19 in (**f**) tRA-treated or (**g**) UV-irradiated (10 J/m$^2$) MEF nuclear extracts w/o RNH and in (**h**) tRA-treated siScrb or si1*Xab2* MEF nuclear extracts. **i** Immunofluorescence detection of γH2AX and 53BP1 (white arrowheads) in wt. MEFs cultured w/o fetal bovine serum (FBS) and transfected (RNH+) or not (RNH-) with RNaseH1. (+FBS ∓RNH: $n = 1070/774$ cells, −FBS ∓RNH: $n = 785/831$ cells/3 independent experiments) (**j**). Native ChIP followed by DRIP on the *Ghr* and *Igf1* promoters in P15 wt. and *Csb^{m/m}* livers. **k** DNA damage inhibits RNA synthesis. XAB2 is released from RNA targets leading to aberrant intron retention and R-loop accumulation. XAB2 interacts with ERCC1-XPF and XPG and the complex is recruited on RNA:DNA hybrids, linking R-loop processing with the spliceosomal response to DNA lesions. DRIP signals are shown, as fold enrichment (F.E) of % input of antibody over % input of control antibody (IgG). The images shown in Fig. 7e–h are representative of experiments that were repeated three times. All scatter and bar blots in this manuscript are presented as mean ± SEM. *p* values were calculated by two-tailed Student's *t* test, unless otherwise indicated. Error bars indicate SEM among three biological replicates in all cases, unless otherwise stated. Source data are provided as a Source Data file.

of each ΔPSI was set as the ratio between the number of differences that are greater than zero and the total number of differences, reflecting the probability of $PSI_{Xab2\ siRNA}$ being greater than $PSI_{Control}$. Differentially spliced events were considered as those with a probability of a $|\Delta PSI| > 0 > 0.8$ and an absolute ΔPSI g > 5% (Table EV3). To assess the enrichment of differentially spliced events in positive or negative ΔPSI values within exon skipping and intron retention events, we tested the null hypothesis that the proportion of positive ΔPSI values (i.e., $PSI_{Xab2\ siRNA} > PSI_{Control}$) was equal to 0.5 using R function *prop.test* that implements the Pearson's chi-squared proportion test.

**RIP studies**. RIP in HEPA cells and MEFs was performed as previously described[87] with a few modifications. In brief, cells ($4 \times 150$ mm plate) were harvested by trypsinization and the cell pellet was resuspended in NP-40 lysis buffer for 10 min on ice. Nuclei were washed once in NP-40 Lysis buffer and subsequently resuspended in 1 ml RIP buffer (150 mM KCl, 25 mM Tris pH 7.4, 5 mM EDTA, 0.5 mM DTT, 0.5% NP40, 1 mM PMSF, 40 U/mL RNaseOut; Invitrogen). Resuspended nuclei were mechanically sheared using a syringe (26 G) with 5-7 strokes. Nuclear membranes and debris were pelleted by centrifugation at $15,800 \times g$ for 10 min. Antibodies (5 μg) were added to the supernatant and incubated overnight at 4 ºC with gentle rotation. Fifty microliters of protein G Sepharose beads (Millipore) were added and incubated for 2 h at 4 ºC with gentle rotation. Beads were pelleted at $3500 \times g$ for 3 min, the supernatant was removed, and beads were washed three times in 1 mL wash buffer (280 mM KCl, 25 mM Tris pH 7.4, 5 mM EDTA, 0.5 mM DTT, 0.5% NP40, 1 mM PMSF, 40 U/mL RNaseOut; Invitrogen) for 10 min at 4 ºC, followed by one wash in PBS. Beads were resuspended in 1 ml of Trizol and co-precipitated RNAs were isolated according to the manufacturer's protocol. RNA was precipitated with Ethanol/Sodium acetate in the presence of Glycoblue at −20 ºC overnight. Isolated RNA was treated with DNase I (Promega) followed by reverse transcription with random primers (Invitrogen). For RIP in liver tissue, livers from two P15 mice were minced and subsequently cross-linked with 0.1% formaldehyde for 10 min at room temperature. After addition of 0.25 M glycine for 5 min, cells were harvested and lysed with RIPA buffer (50 mM Tris-HCl [pH 7.4], 1% NP-40, 0.5% Na deoxycholate, 0.05% SDS, 1 mM EDTA, and 150 mM NaCl) followed by sonication at 4 °C. Nuclear membrane and debris were pelleted by centrifugation at $15,800 \times g$ for 10 min. Antibodies (5 μg) were added to the supernatant and incubated overnight at 4 ºC with gentle rotation. Fifty microliters of protein G Sepharose beads (Millipore) were added and incubated for 2 h at 4 ºC with gentle rotation. Beads were pelleted at $3500 \times g$ for 3 min, the supernatant was removed, and beads were washed three times in 1 mL wash buffer (50 mM Tris-HCl [pH 7.4], 1% NP-40, 0.5% Na deoxycholate, 0.05% SDS, 1 mM EDTA, and 350 mM NaCl) for 10 min at 4 ºC, followed by one wash with PBS. Crosslinks were reversed by adding 100 μl elution buffer (50 mM Tris-HCl pH 6.5, 5 mM EDTA, 1% SDS, and 10 mM DTT) and heating for 45 min at 70 °C. RNA was purified with TRIzol reagent, treated with DNase I, and used for first-strand cDNA synthesis.

**RNA-Seq and quantitative PCR studies**. Total RNA was isolated from cells using a Total RNA isolation kit (Qiagen) as described by the manufacturer. For RNA-Seq studies, libraries were prepared using the Illumina® TruSeq® mRNA stranded sample preparation Kit for HEPA cells and for mESCs. Library preparation started with 1 μg total RNA. After poly-A selection (using poly-T oligo-attached magnetic beads), mRNA was purified and fragmented using divalent cations under elevated temperature. The RNA fragments underwent reverse transcription using random primers. This is followed by second-strand cDNA synthesis with DNA polymerase I and RNase H. After end repair and A-tailing, indexing adapters were ligated. The products were then purified and amplified to create the final cDNA libraries. After library validation and quantification (Agilent 2100 Bioanalyzer), equimolar

amounts of all 12 libraries were pooled. The pool was quantified by using the Peqlab KAPA Library Quantification Kit and the Applied Biosystems 7900HT Sequence Detection System. The pool was sequenced by using an Illumina HiSeq 4000 sequencer with a paired-end ($2 \times 75$ cycles) protocol. Quantitative PCR (Q-PCR) was performed with a Biorad 1000-series thermal cycler according to the instructions of the manufacturer (Biorad) as previously described[31]. All relevant data and primer sequences for the genes tested by qPCR are available upon request.

**DRIP**. DRIP analysis was based on ChIP analysis with some modifications. DRIP analysis was performed without a cross-linking step. Nuclei were isolated using 0.5% NP-40 buffer. Isolated nuclei were resuspended in TE buffer supplemented with 0.5%SDS and 100mg proteinase K. Genomic DNA was isolated after the addition of potassium acetate (1 M) and isopropanol precipitation. DNA was sonicated on ice 3 min using Covaris S220 Focused ultrasonicator. Samples were treated with RNase H (10 units/5 μg DNA) at 37 ℃ overnight. Samples were immunoprecipitated with S9.6 antibodies (8 μg antibody/5 μg DNA) overnight at 4 ℃ followed by incubation for 3 h with protein G-Sepharose beads (Millipore) and washed sequentially. The complexes were eluted and purified DNA fragments were analyzed by qPCR using sets of primers targeting different regions of related genes.

**DRIP western analysis**. DRIP western analysis was performed as described previously[78]. Briefly, non-crosslinked cells were lysed in 0.5% NP40 buffer for 10 min on ice. Pelleted nuclei were lysed in RSB buffer (10 mM Tris-HCl pH 7.5, 200 mM NaCl, 2.5 mM MgCl2) with 0.2% sodium deoxycholate [NaDOC, 0.1% SDS and 0.5% Triton X-100, and extracts were sonicated for 10 min (Diagenode Bioruptor). Extracts were then diluted 1:4 in RSB with 0.5% Triton X-100 (RSB + T) and subjected to IP with the S9.6 antibody (8 μg antibody/ 5 μg DNA), bound to protein A dynabeads (Invitrogen), and pre-blocked with 1 mg/ml BSA/PBS for 1 h. IgG2a antibodies were used as control. RNase H (PureLink, Invitrogen) was added before IP as in DRIP. Beads were washed 4× with RSB + T; 2× with RSB; and eluted in 1× Laemmli.

**Sequential native ChIP analysis**. Non-cross-linked cells were lysed using 0.5% NP-40 buffer. Chromatin was digested with MNase (50 Units/0.5mgDNA) at 37 ℃ for 10 min. S1 chromatin fraction was isolated. S2 chromatin fraction was dialyzed against Tris-EDTA buffer for 2 h and isolated. S1 and S2 chromatin fractions were used for pulldown using M280 paramagnetic streptavidin beads (Invitrogen). After sequential washes complexes were eluted. A fraction of them was kept for qPCR analysis and the rest was used for S9.6 IP as described above.

**Chromatin associated RNA assay (CAR)**. Cells are lysed [0.15% (vol/vol) NP-40, 10 mM Tris-HCl (pH 7.0), 150 mM NaCl, 10 U RNase inhibitor, 1x Protease inhibitor mix] and the cytoplasm is separated from nuclei by centrifugation through a sucrose cushion (10 mM Tris-HCl (pH 7.0), 150 mM NaCl, 25% sucrose). The nuclei pellet is washed to remove cytoplasmic remnants prior to chromatin fractionation. The chromatin is separated from the nucleoplasm in the presence of Urea, the nonionic detergent Nonidet P-40 (NP-40) and NaCl [1% (vol/vol) NP-40, 20 mM HEPES (pH 7.5), 300 mM NaCl, 1 M urea, 0.2 mM EDTA, 1 mM DTT, 10 RNase inhibitor, 1× Protease inhibitor mix]. as previously described[88] Next, the RNA is prepared from the chromatin fractions and the remaining DNA is removed by a standardized DNaseI digest.

**Gel mobility shift assays**. The XAB2 complex or recombinant *Xab2* (0,1-1-5-10 nmoles; Abnova) was incubated with the biotinylated RNA probes (100 pmoles) in binding buffer (10 μl) containing 10 mm Tris-HCl (pH 8.0), 50 mm KCl, 10%

glycerol, 0.5 mm EDTA, 0.5 mm dithiothreitol, 1 µg/ml BSA and 6 mm MgCl₂ for 30 min at 4 °C as previously described[20]. After the incubation, the binding reaction mixtures are separated on a non-denatured 4% polyacrylamide gel in 0.5× TBE buffer, pH:7,5. Complexes and free oligos are transferred on a nylon membrane and UV cross-linked. The presence of biotin was detected with stp-HRP.

**Primer sequences**. Primer sequences are provided in Supplementary Table 1.

**Quantification and statistical analysis**. A two-tailed *t*-test was used to extract the statistically significant data by means of the IBM SPSS Statistics 19 (IBM) and the *R* software for statistical computing (www.r-project.org). Significant over-representation of pathways and gene networks was determined by GO (http://geneontology.org/). Data analysis is discussed also in the Method Details section. Experiments were repeated at least three times. The data exhibited normal distribution (where applicable). There was no estimation of group variation before experiments. Error bars indicate standard deviation unless stated otherwise (standard error of the mean; s.e.m.). For animal studies, each biological replicate consists of 3–5 mouse tissues or cell cultures per genotype per time point or treatment. No statistical method was used to predetermine sample size. None of the samples or animals was excluded from the experiment. The animals or the experiments were non-randomized. The investigators were not blinded to allocation during experiments and outcome assessment.

**Reporting summary**. Further information on research design is available in the Nature Research Reporting Summary linked to this article.

## Data availability
The mass spectrometry proteomics data have been deposited to the ProteomeXchange Consortium (http://proteomecentral.proteomexchange.org) via the PRIDE partner repository with the dataset identifier PXD014084. The RNA-Seq data are deposited in ArrayExpress (https://www.ebi.ac.uk/arrayexpress/) under accession code E-MTAB-8035. Source data are provided with this paper. All other data and reagents are available from the authors upon reasonable request. Source data are provided with this paper.

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

## Acknowledgements

The Horizon 2020 ERC Consolidator grant "DeFiNER" (GA64663), the Horizon 2020 Marie Curie ITNs "Chromatin3D" (GA622934), "aDDRess" (GA812829) and "HealthAge" "(GA812830), the Hellenic Foundation for Research and Innovation (HFRI) and the General Secretariat for Research and Technology (GSRT) under grant agreement HFRI-1059 and HFRI-FM17-631 and the "Fondation Santé" grant supported this work. G.C. is supported by the IKY postdoctoral research fellowship program (MIS: 5001552), co-financed by the European Social Fund- ESF and the Greek government. N.L.B.-M. is supported by an EMBO Installation Grant (3057) and an Investigador FCT (Fundação para a Ciência e a Tecnologia) Starting Grant (IF/00595/2014); M.A.-F. is supported by an FCT PhD fellowship (PD/BD/128283/2017) and Fundação AstraZeneca.

## Author contributions

E.G., M.T., N.B., M.A.-F., E.L., K.S., G.C., P.T., T.K., J.A., J.A.D., and N.L.B.-M. performed the experiments and/or analyzed data. G.A.G. interpreted data and wrote the manuscript. All relevant data are available from the authors.

## Competing interests

The authors declare no competing interests.
