## [Peer Review File · Nature Communications]

REVIEWER COMMENTS

Reviewer #1 (Remarks to the Author):

This study provides novel mechanistic insight into the function of Xab2, a protein previously known to be involved in splicing and transcription coupled repair. Authors generated a very useful tool to explore the binding partners of XAB2 in mouse cells by fusing the protein with a tandem affinity purification tag. Proteomic analysis of XAB2-associated complexes was done in P15 livers and XAB2-bound proteins before and after UV-irradiation were identified in MEFs. Knockdown experiments were further performed in HEPA cells.

1) In the title, abstract and throughout the text, authors claim a role for XAB2 in mammalian development. This conclusion is not supported by the experimental data, which is collected from either P15 livers, MEFs and HEPA.

2) Fig. 4, more controls are needed to conclude that XAB2 binds directly to UsnRNA, namely the binding sequence in the RNA should be identified. Similar results could be obtained if XAB2 associated with snRNP protein components, not necessarily via the snRNA. Indeed, Xab2 is not predicted to contain RNA binding domains, therefore it is unlikely that it binds directly to RNA.

3) Line 257/258, it is not correct to conclude that "XBA2 is required for proper mRNA splicing early on during mammalian development" based only on KD experiments in HEPA cells.

4) Fig. 5 should include data on U5snRNA as well as U4 and U6 snRNP proteins – are these components also displaced upon UV treatment? The results are consistent with a general disassembly of the spliceosome. If that is the case there is no specific involvement of XBA2.

5) Line 283 - Authors state: "indicating that the presence of DNA helix-distorting lesions inadvertently affects the process of pre-mRNA splicing in the mammalian genome" – figure 3C shows a delay in the recovery of RNA synthesis upon UV treatment, this suggests lesions may displace transcription and all transcriptional associated processes. In addition to splicing inhibition, the XAB2 interatome should be analysed after transcriptional inhibition.

6) Fig. 6, knockdown of XAB2 triggers the formation of R-loops. However, this may be an indirect effect secondary to splicing inhibition. Is the role of XAB2 in R-loop metabolism independent from splicing?

Minor points:

Fig2E - The red line not visible, FDR FE GC (bring to E) ,

Fig2G - Missing colour legends; what are the black coloured proteins?

Fig3A - Missing staining information

Fig3D and E – Missing statistical analysis

Line 333 - Authors refer to (Figure 6A-B) but they mean (Figure 5A-B) instead

Fig 6A - missing untreated UV control

Reviewer #2 (Remarks to the Author):

This is a review of Manuscript#: NCOMMS-20-29014. The protein XAB2 has previously been linked to nucleotide excision repair (NER), the spliceosome, and other aspects of DNA repair (e.g. homologous recombination) and RNA transcription/processing. However, the links between these functions remain poorly understood. This manuscript provides interesting new links between XAB2 function in NER and splicing. The main approach is immunoprecipitation of XAB2 complexes using tagging of the endogenous protein in mice, with most of the analysis in nuclear extracts from the liver. The first few figures validate this approach & provide comprehensive analysis of XAB2 protein complexes, as well as XAB2-RNA interactions. Furthermore, with siRNA targeting of XAB2, its role in promoting NER, limiting DNA damage, and limiting R-loop accumulation are confirmed. These findings are comprehensive/compelling, and largely confirm the literature about this protein, albeit using endogenous protein from primary cells. Then, in Figures 5 & 7, the links between the NER and

splicing function of XAB2 are probed, finding that UV-induced DNA damage causes loss of XAB2 (and associated factor AQR) from the spliceosome (probed as interaction with U4/U6 snRNAs). Through multiple methods, this displacement of XAB2 from U4/U6 is shown to be dependent on DNA damage, as it is suppressed via efficient repair of the DNA lesions. Then, in Figures 7E/F, the association of XAB2 with the NER factor XPF is shown to be inhibited by RNASEH treatment. Similarly, pulldown of R-loops can isolate the XAB2-XPF(XPG) complex in a manner that is disrupted by RNASEH. Altogether, these findings support a model whereby DNA damage causes redistribution of XAB2 away from the spliceosome, but association with NER factors at R-loops. These findings are significant not only for understanding XAB2 function, but provide a roadmap for studies on the interplay between spliceosome regulation and the DNA damage response / R-loop metabolism.

Major point:

In my opinion, a gap is whether the association of XAB2 w/ R-loops and XPF is affected by UV-induced DNA damage. Namely, regarding 7D / E, if cells are treated with both tRA (to induce transcription / induced XAB2-R-loop interaction / XAB2-XPF S9.6 pulldown) and UV, does UV treatment enhance / suppress / or have no effect on these interactions? Perhaps this is within supplemental data, and I didn't find it. I think this is important, because it influences the conclusion, regarding the effect of DNA damage on the interplay between spliceosome regulation and R-loop regulation. Namely, since UV treatment cause loss of XAB2 interaction with the splicesome, does this allow enhanced association of XAB2 with R-loops/NER, or is the interaction of XAB2 with R-loops/NER independent of DNA damage (or even suppressed)?

Minor points:

Some of the combined / two-step IP experiments (Fig 7D), and the critical Fig 7E-H could be described in greater detail in the Results. For example, in the Results section, the description of Fig 7 E/F seemed jumbled together, where they are different experiments.

Overall, the Results could have more information/clarity on what is confirmatory from the literature, rather than just adding the citations at the end of particular sentences. Namely, some more sentences like "Similar to previous findings" or something. As an example, the XAB2 - gH2AX adjacent localization has already been reported (PMID: 27084940). Overall, the description of the prior literature is solid, but I think such sentences in the Results will clarify what is confirmatory (or maybe contradictory), which also enhances/emphasizes the novel findings of this study.

Reviewer #3 (Remarks to the Author):

Title: The splicing factor XAB2 interacts with 2 ERCC1-XPF and XPG for RNA-loop 3 processing during mammalian development

Authors: Evi Goulielmaki et al

The manuscript by Goulielmaki and co-authors describes an effort to characterize the role of XAB2 in RNA-loop processing during mammalian development.

Comments:

Introduction

I really appreciate the detailed splicing initiation mechanism in the introduction. Authors did a great job to explain the problem with studying the functional role of XAB2 in DDR in vivo and introduce the approach of in vivo biotinylation tagging. Also, the last paragraph of the introduction where the

authors are talking about their findings in details, should be modified and shortened, as this should be highlighted later in the discussion. Authors should perhaps explain here a little bit more their approach, as this approach led to their results. When authors say 'using a series of immunoprecipitation approaches' I am left confused, because it is unclear to me what this actually means, so I jumped to methods section.

Methods

In general, methods are written in a confusing manner, and are hard to follow in order to understand the strategy.

For the mass spectrometry studies, that are crucial for the findings in this manuscript, it is necessary to make things clean, clear and simple in the Methods section. There is no information here about what was the starting amount of proteins or beads. Moreover, there is no explanation what samples were used here, how many replicates, what was the negative control? In the interactome analysis and interpretation, there has to be well defined negative control, and authors should give details about it in the Methods section. If this is discussed in the Results section, it also needs to be included here. I don't understand why the in-gel digestion strategy was used here instead of the in-solution. Authors should explain their decisions and give more details about how the gel lanes were cut/combined. In the row 634, author should omit the word 'directly' sprayed, as this was not direct injection and hence is the wrong terminology. I suggest that Mass Spectrometry studies is broken into 2 parts: LC-MS/MS analysis and bioinformatics.

Co-IP method: authors should describe here their affinity-tag strategy for IP for mass spec that should be put before mass spectrometry part, as here we see what the amount of proteins used was. Please provide the amount of beads used for mass spectrometry purposes. Please clearly describe what exactly the method here was for mass spectrometry, as it is not clear when talking about '15-day old livers, cells, normal mouse IgG, rabbit IgG'. I am really struggling to understand what was done here for the mass spec experiment. Please also omit the word 'normal' when describing IgG. Were these IgG used as a negative control for IPs? Using them is a good way to catch 'sticky' proteins that are not necessarily interactors.

Results

Fig2A. Please modify the description, as this figure do not represent the high-throughput MS analysis. It rather defines the starting material, and products after Co-IP. This would be an ideal space to make a detailed scheme of your whole Co-IP coupled MS approach. Please better define BiRA mice, so it is clear why this is your control for the MS experiment. I would appreciate if you used 'IP' instead of 'PD' in your figures, so it matches the terminology used in Methods section. Please explain here why is PRP19 important for your IPs. Fig2C. having MW markers marked here is absolutely irrelevant, in contrast to marking how many replicates were run, and saying somewhere that these are representative samples, also marking negative control. Again, I believe that detailed schematic representation of the whole affinity-tagged MS is necessary.

The part with XPF proteome introduction and Fig 2G are very confusing, and needs to be re-written.

Nature Communications NCOMMS-20-29014: "The Splicing Factor XAB2 interacts with ERCC1-XPF and XPG for RNA-loop processing"

Authors' reply to reviewers' remarks

We would like to thank the reviewers for their valuable comments in our work. We have carefully evaluated their comments and suggestions and performed several additional experiments to further strengthen the data and the conclusions presented in our manuscript. Please find below our point-by-point response to the Reviewers remarks. Text changes in the manuscript are highlighted in light grey color.

Reviewers' Comments:

Reviewer #1 (Remarks to the Author):

Reviewer's remark: *This study provides novel mechanistic insight into the function of Xab2, a protein previously known to be involved in splicing and transcription coupled repair. Authors generated a very useful tool to explore the binding partners of XAB2 in mouse cells by fusing the protein with a tandem affinity purification tag. Proteomic analysis of XAB2-associated complexes was done in P15 livers and XAB2-bound proteins before and after UV-irradiation were identified in MEFs. Knockdown experiments were further performed in HEPA cells.*

Authors' reply: We would like to thank the Reviewer for her/his supportive remarks.

Reviewer's remark 1: *In the title, abstract and throughout the text, authors claim a role for XAB2 in mammalian development. This conclusion is not supported by the experimental data, which is collected from either P15 livers, MEFs and HEPA.*

Authors' reply: We agree with the Reviewer. Our work provides data on the functional role of XAB2 in pre-mRNA splicing, DNA repair and R-loop processing in developing P15 livers, mESCs and primary MEFs or HEPA cells but not during mammalian development. The manuscript title now reads: "The Splicing Factor XAB2 interacts with ERCC1-XPF and XPG for RNA-loop processing". The term "mammalian development" has also been removed from the main text body of the manuscript.

Reviewer's remark 2: *Fig. 4, more controls are needed to conclude that XAB2 binds directly to UsnRNA, namely the binding sequence in the RNA should be identified. Similar results could be obtained if XAB2 associated with snRNP protein components, not necessarily via the snRNA.*

Indeed, Xab2 is not predicted to contain RNA binding domains, therefore it is unlikely that it binds directly to RNA.

Authors' reply: We thank the Reviewer for this excellent remark. Indeed, a new set of data in the revised manuscript reveal that XAB2 does not bind directly to UsnRNAs. Specifically, to test this, we first performed a series of gel mobility shift assays using a biotinylated RNA oligo that was designed based on the sequence similarities of U4, U5 and U6 snRNAs with the minimal RNA binding sequence of HCF107, a chloroplast-localized protein with repeats of HAT motifs (**Figure S1L**). This strategy reveals that the XAB2 IP eluent from nuclear extracts of MEFs consumes the RNA oligo but not its mutant version (**Figure S1M**) indicating that the XAB2 complex preferentially binds to the UsnRNA-specific sequence. Next, to test whether XAB2 binds directly to the same RNA sequence, we incubated increasing amounts of the recombinant *hnXAB2* with the biotinylated RNA oligo (**Figure S1N**). This approach revealed no consumption of the RNA oligo. To accommodate the new findings, the term “binds to UsnRNAs” was replaced with “is in complex with” or “associates with UsnRNAs” throughout the manuscript text.

Reviewer's remark 3: *Line 257/258, it is not correct to conclude that “XBA2 is required for proper mRNA splicing early on during mammalian development” based only on KD experiments in HEPA cells.*

Authors' reply: In addition to the knockdown experiments (KD) in HEPA cells, RNA-Seq analysis was performed in XAB2 KD mESCs as well (**Figure 4G-H**) showing also in this case a global impairment of the splicing machinery in *siXab2* cells. However, we agree with the Reviewer's comment and have removed the term “mammalian development” from the manuscript text. The sentence now reads “Thus, XAB2 is part of the core spliceosome complex and has a functional role in pre-mRNA splicing; it is recruited on UsnRNAs and pre-mRNAs and it is required for proper mRNA splicing”.

Reviewer's remark 4: *Fig. 5 should include data on U5snRNA as well as U4 and U6 snRNP proteins – are these components also displaced upon UV treatment? The results are consistent with a general disassembly of the spliceosome. If that is the case there is no specific involvement of XBA2.*

Authors' reply: We provide a new set of data regarding the association of XAB2 with U5 snRNA upon UV irradiation and a new series of RNA immunoprecipitation (RIP) data with U4/U6 snRNP PRP3. XAB2 and PRP3 RIP signals (**Figure 5A** and **Figure S3G**) together with a series of Chromatin Associated RNA assays for U4 and U6 snRNAs (**Figure S3F**) and a BrU incorporation assay to evaluate the recovery of RNA synthesis (**Figure S4B**) indicate XAB2 and the core spliceosome are displaced within 2 hours post-UV irradiation. However, we also find that the core

spliceosome re-assembles 6 hours after UV exposure when XAB2 association with UsnRNAs remains significantly reduced (**Figure 5A**).

Reviewer's remark 5: *Line 283 - Authors state: "indicating that the presence of DNA helix-distorting lesions inadvertently affects the process of pre-mRNA splicing in the mammalian genome" – figure 3C shows a delay in the recovery of RNA synthesis upon UV treatment, this suggests lesions may displace transcription and all transcriptional associated processes. In addition to splicing inhibition, the XAB2 interatome should be analysed after transcriptional inhibition.*

Authors' reply: In addition to the splicing inhibition, we now provide a new series of data showing that upon inhibition of transcription initiation (TPL), transcription elongation (DRB) or splicing (isoG), the association of XAB2 with U4/6 snRNAs (**Figure S4C-D**) or with all pre-mRNAs tested (in the case of TPL or DRB treatments) (**Figure S4E**) is significantly reduced.

Reviewer's remark 6: *Fig. 6, knockdown of XAB2 triggers the formation of R-loops. However, this may be an indirect effect secondary to splicing inhibition. Is the role of XAB2 in R-loop metabolism independent from splicing?*

Authors' reply: A new series of ChIP-DRIP show that, upon transcription induction, XAB2 is recruited on R-loops that accumulate on the promoter regions of tRA-regulated, intronless *Fibin* and *Sstr4* genes (**Figure S5F**). Importantly, R-loops further accumulate in the promoters of intronless *Fibin* and *Sstr4* genes when XAB2 is released from these gene targets (**Figure S5B** and **Figure S5B**) in tRA/UV-treated MEFs.

Minor points:

Reviewer's remark: *Fig2E - The red line not visible, FDR FE GC (bring to E),*

Authors' reply: This has now been corrected.

Reviewer's remark: *Fig3A - Missing staining information*

Authors' reply: This has now been corrected.

Reviewer's remark: *Fig3D and E – Missing statistical analysis*

Authors' reply: This has now been corrected.

Reviewer's remark: *Line 333 - Authors refer to (Figure 6A-B) but they mean (Figure 5A-B) instead*

Authors' reply: This has now been corrected.

Reviewer's remark: *Fig 6A - missing untreated UV control*

Authors' reply: The data shown represent the fold change to untreated UV control sample.

Reviewer's remark: *Fig2G - Missing colour legends; what are the black coloured proteins?*

Authors' reply: We apologize for the inconvenience. The black colored proteins represent the protein components of the TREX (transcription and export)/ECJ (exon junction) complex.

Reviewer #2 (Remarks to the Author):

Reviewer's remark: *These findings are significant not only for understanding XAB2 function, but provide a roadmap for studies on the interplay between spliceosome regulation and the DNA damage response /R-loop metabolism.*

Authors' reply: We would like to thank the Reviewer for her/his supportive remarks.

Reviewer's remark 1: *Major point: In my opinion, a gap is whether the association of XAB2 w/ R-loops and XPF is affected by UV-induced DNA damage. Namely, regarding 7D / E, if cells are treated with both tRA (to induce transcription / induced XAB2-R-loop interaction / XAB2-XPF S9.6 pulldown) and UV, does UV treatment enhance / suppress / or have no effect on these interactions? Perhaps this is within supplemental data, and I didn't find it. I think this is important, because it influences the conclusion, regarding the effect of DNA damage on the interplay between spliceosome regulation and R-loop regulation. Namely, since UV treatment cause loss of XAB2 interaction with the spliceosome, does this allow enhanced association of XAB2 with R-loops/NER, or is the interaction of XAB2 with R-loops/NER independent of DNA damage (or even suppressed)?*

Authors' reply: We thank the Reviewer for raising this important question. To test whether XAB2 interaction with R-loops is affected in tRA-treated cells also exposed to UV irradiation, we performed a set of sequential XAB2 ChIP-DRIP experiments on tRA-induced gene promoters (**Figure S5E-F**), previously shown to accumulate R-loops upon tRA/UV treatment (**Figure S5B**). We find that XAB2 is released from R-loops in tRA/UV-treated cells. XAB2 is also released from R-loops in tRA/UV-treated bXAB2;*Csb^{m/m}* MEFs (that are defective in TC-NER) and in tRA/UV-treated bXAB2;*Xpc^{-/-}* MEFs (that are defective in GG-NER) (**Figure S5G**). The release of XAB2 from R-loops upon DNA damage is also evident by DRIP-western analysis in the nuclear extracts of UV-irradiated MEFs (**Figure 7G**) as well as in tRA/UV-treated MEFs (**Figure S6H**). Lastly, a series of bXAB2 (**Figure S6F**) and bXPF (**Figure S6D-E**) pulldowns revealed that the interaction of XAB2 with XPF and XPG is lost in tRA/UV-treated MEFs. Taken together, our findings indicate

that, upon UV irradiation, XAB2 is released from R-loops in tRA-treated cells also exposed to UV irradiation highlighting the impact of persistent DNA damage on R-loop processing.

Minor points:

Reviewer's remark: *Some of the combined / two-step IP experiments (Fig 7D), and the critical Fig 7E-H could be described in greater detail in the Results. For example, in the Results section, the description of Fig 7 E/F seemed jumbled together, where they are different experiments.*

Authors' reply: To enhance the readability of the manuscript, we have now rephrased the text and altered the sequence of Figure 7E and Figure 7F.

Reviewer's remark: *Overall, the Results could have more information/clarity on what is confirmatory from the literature, rather than just adding the citations at the end of particular sentences. Namely, some more sentences like "Similar to previous findings" or something. As an example, the XAB2 - γ H2AX adjacent localization has already been reported (PMID: 27084940). Overall, the description of the prior literature is solid, but I think such sentences in the Results will clarify what is confirmatory (or maybe contradictory), which also enhances/emphasizes the novel findings of this study.*

Authors' reply: In several instances, we have now clarified the findings, which are confirmatory to previous published work further enhancing the novel data presented in our manuscript.

Reviewer #3 (Remarks to the Author):

Reviewer's remark: *I really appreciate the detailed splicing initiation mechanism in the introduction. Authors did a great job to explain the problem with studying the functional role of XAB2 in DDR in vivo and introduce the approach of in vivo biotinylation tagging. Also, the last paragraph of the introduction where the authors are talking about their findings in details, should be modified and shortened, as this should be highlighted later in the discussion. Authors should perhaps explain here a little bit more their approach, as this approach led to their results. When authors say 'using a series of immunoprecipitation approaches' I am left confused, because it is unclear to me what this actually means, so I jumped to methods section.*

Authors' reply: We would like to thank the Reviewer for her/his supportive remarks. The last paragraph has been shortened. The *in vivo* biotinylation approach is described in great detail in the first section of the results "Generation of biotin-tagged XAB2 mice".

Reviewer's remark: *For the mass spectrometry studies, that are crucial for the findings in this manuscript, it is necessary to make things clean, clear and simple in the Methods section. There is no information here about what was the starting amount of proteins or beads. Moreover, there is no*

explanation what samples were used here, how many replicates, what was the negative control? In the interactome analysis and interpretation, there has to be well defined negative control, and authors should give details about it in the Methods section. If this is discussed in the Results section, it also needs to be included here. I don't understand why the in-gel digestion strategy was used here instead of the in-solution. Authors should explain their decisions and give more details about how the gel lanes were cut/combined. In the row 634, author should omit the word 'directly' sprayed, as this was not direct injection and hence is the wrong terminology. I suggest that Mass Spectrometry studies is broken into 2 parts: LC-MS/MS analysis and bioinformatics.

Authors' reply: To satisfy the reviewer's remark, we have now split the description of the mass/spec studies into two parts i.e. 1. the Mass Spectrometry (MS) studies and 2. the LC-MS/MS data analysis. The mass spectrometry section describes in detail the pulldown protocol along with information on the amount of the starting material and the beads, the samples used, the number of replicates and the negative control used. The 'in gel' strategy was chosen over the in-solution approach to allow the possibility to verify the molecular weight of the identified proteins, thereby minimizing any false positive hits. This information is now provided in the text further clarifying how gel lanes were cut.

Reviewer's remark: *Co-IP method: authors should describe here their affinity-tag strategy for IP for mass spec that should be put before mass spectrometry part, as here we see what the amount of proteins used was. Please provide the amount of beads used for mass spectrometry purposes. Please clearly describe what exactly the method here was for mass spectrometry, as it is not clear when talking about '15-day old livers, cells, normal mouse IgG, rabbit IgG'. I am really struggling to understand what was done here for the mass spec experiment. Please also omit the word 'normal' when describing IgG. Were these IgG used as a negative control for IPs? Using them is a good way to catch 'sticky' proteins that are not necessarily interactors.*

Authors' reply: A co-IP strategy has not been used in the present work. The *in vivo* biotinylation tagging and pulldown strategies are described in the first section of the results "Generation of biotin-tagged XAB2 mice" and in the methods section "Mass Spectrometry (MS) studies". We have also omitted the word "normal" when referring to the IgG serum control used in our IP experiments.

Reviewer's remark: *Fig2A. Please modify the description, as this figure do not represent the high-throughput MS analysis. It rather defines the starting material, and products after Co-IP. This would be an ideal space to make a detailed scheme of your whole Co-IP coupled MS approach. Please better define BiRA mice, so it is clear why this is your control for the MS experiment. I would appreciate if you used 'IP' instead of 'PD' in your figures, so it matches the terminology used in Methods section. Please explain here why is PRP19 important for your IPs. Fig2C. having MW markers marked here is absolutely irrelevant, in contrast to marking how many replicates were run, and saying somewhere that these are representative samples, also marking negative control. Again, I believe that detailed schematic representation of the whole affinity-tagged MS is necessary.*

Authors' reply: We have further adapted the schematic representation of the *in vivo* biotinylation tagging, isolation and mass spectrometry strategy in Figure 2A. We have also modified the text appropriately to further clarify the use of BirA mouse liver extracts (as the negative controls used in our work) and the relevance of PRP19 Western blotting that was used to validate the pull-down strategy. We have also modified the methods that now clearly distinguish the use of an affinity tag (pulldowns) from an antigen-antibody (immunoprecipitation) interaction.

Reviewer's remark: The part with XPF proteome introduction and Fig 2G are very confusing, and needs to be re-written.

Authors' reply: We apologize for this inconvenience. We have now added the missing information in Figure 2G legend.

REVIEWERS' COMMENTS

Reviewer #1 (Remarks to the Author):

Authors have carefully addressed all my criticisms. The new experiments added make the conclusions much stronger. In my opinion, the manuscript is ready for publication.

Reviewer #2 (Remarks to the Author):

My concerns have been addressed with excellence.

Reviewer #3 (Remarks to the Author):

I would like to thank authors for addressing all concerns raised from my side, as the manuscript looks in much better shape now.